# Deep learning extends de novo protein modelling coverage of genomes using iteratively predicted structural constraints

Joe G. Greener[1,2,3], Shaun M. Kandathil[1,2,3] & David T. Jones[1,2]

The inapplicability of amino acid covariation methods to small protein families has limited their use for structural annotation of whole genomes. Recently, deep learning has shown promise in allowing accurate residue-residue contact prediction even for shallow sequence alignments. Here we introduce DMPfold, which uses deep learning to predict inter-atomic distance bounds, the main chain hydrogen bond network, and torsion angles, which it uses to build models in an iterative fashion. DMPfold produces more accurate models than two popular methods for a test set of CASP12 domains, and works just as well for transmembrane proteins. Applied to all Pfam domains without known structures, confident models for 25% of these so-called dark families were produced in under a week on a small 200 core cluster. DMPfold provides models for 16% of human proteome UniProt entries without structures, generates accurate models with fewer than 100 sequences in some cases, and is freely available.

[1] Department of Computer Science, University College London, Gower Street, London WC1E 6BT, UK. [2] The Francis Crick Institute, 1 Midland Road, London NW1 1AT, UK. [3]These authors contributed equally: Joe G. Greener, Shaun M. Kandathil. Correspondence and requests for materials should be addressed to D.T.J. (email: d.t.jones@ucl.ac.uk)

I n recent years, the ability to accurately predict residue-residue contacts in protein structures from a family of protein sequences has increased dramatically, mainly due to the recent breakthrough in developing statistical models which can separate direct from indirect correlation effects[1,2]. Methods to generate accurate protein models from predicted contacts, which may be incomplete or have many false positives, have received far less attention. Model generation is usually treated as a separate step from contact prediction[3].

Existing approaches for template-free model generation using predicted residue-residue contacts tend to fall into two categories. Well established fragment-based methods such as Rosetta[4] and FRAGFOLD[5] add constraints from predicted contacts to an existing fragment assembly pipeline. Fragment-based methods have performed well in the biennial Critical Assessment of protein Structure Prediction (CASP) experiments[6] but take a large amount of computing power, produce a variable fraction of native-like models and are dependent on native-like fragments being available. For complex beta-sheet topologies, particularly those with high contact order, fragment assembly often fails to produce a viable model, despite attempts at overcoming these limitations[7].

An attractive, but far less popular approach to de novo modelling has been to use distance geometry to project contact information into 3-D space, similar to the procedures used in NMR structure determination e.g., the DRAGON method of Taylor and Aszódi[8]. More recently, however, distance geometry-based approaches have become more widely used thanks to the improvements in covariation-based contact predictions. For example, CONFOLD2[9] and EVfold[10] add constraints from predicted contacts to standard inter-atomic distance constraints, then use software such as CNS[11] to generate coordinates from these constraints. CONFOLD2 integrates secondary structure and predicted contacts in a two-stage modelling approach, where unsatisfied contacts are filtered out after initial model generation. These methods are computationally cheaper than fragment assembly but produce poor models without a large number of sufficiently accurate predicted contacts. They are also susceptible to producing mirrored topologies where the secondary structures have the correct handedness, but the overall 3-D packing of the secondary structural elements is itself a mirror of the native structure[12,13].

At the recent CASP13 experiment, methods using deep learning approaches to predict distances to use in model building appeared for the first time. AlphaFold used predicted distance likelihoods to generate protein family-specific potentials of mean force that could be directly minimised to generate accurate models. Raptor-X used predicted distances as constraints in CNS to generate models[14]. Two recently proposed deep learning methods attempt to generate model coordinates directly from sequence data by end-to-end training[15,16]. Whilst promising, these end-to-end trained methods have not yet shown anything close to state-of-the-art performance in protein modelling, probably because they do not make use of the recent advances in sequence covariation analysis.

Here, we introduce DMPfold, a development of our Deep-MetaPSICOV (DMP) contact predictor[17]. Rather than predicting contacts, DMPfold predicts inter-atomic distance bounds, torsion angles and hydrogen bonds and uses these constraints to build models. An iterative process of model generation and constraint refinement is used to filter out unsatisfied constraints. Other modifications to the neural network architectures also differentiate DMPfold from DMP; see the Methods section. We show that DMPfold produces more accurate models than CONFOLD2 and Rosetta for the CASP12 free modelling (FM) domains, with particularly good performance when asked to generate just a single best model. It can also produce high quality models for a set of biologically important transmembrane proteins without any modification for the different prediction task.

Validating the accuracy of a protein structure prediction method is necessary, but an important follow-up question, of more interest to the wider biological research community, is how useful these new methods are in practical terms. In order to demonstrate the utility of DMPfold, we run it on Pfam[18], a database of protein families. The advantage of this is that it allows these de novo models to be mapped easily to whole genomes so that genome-wide coverage can be assessed. A number of previous studies have also done this[13,19,20], so this represents a good indication of how effective deep learning is on an important structural biology problem. First, we show the accuracy of DMPfold on a validation set of over 1000 Pfam families with structures that are known, but not similar to those used in the training of DMPfold. Second, we use this set to develop and verify an accurate estimator of final model accuracy. Finally, we provide to the community models for 1475 Pfam families that are currently lacking structures. Using our predictor of model accuracy, 83% of these models are predicted to have the correct fold. We estimate the increase this provides in the structural coverage of various organisms. This consolidates recent advances in protein structure prediction into a freely available tool that takes just a few hours to run on a standard desktop computer for a typical 200 residue protein domain, making high quality protein structure prediction readily available to the research community. The source code, documentation, trained neural network models and Pfam 3-D models are available at https://github.com/psipred/DMPfold.

## Results

**DMPfold on CASP12 targets**. This work focuses on the ability of DMPfold to produce accurate models and to carry out genome-scale structure prediction. The ability of DMP to predict residue-residue contacts is assessed separately[17]. In order to compare DMPfold to existing model generation methods that utilise contacts, we convert predicted distances from DMPfold to predicted contacts by summing up the likelihoods in distance bins up to 8 Å. These contacts are then used to generate models with CONFOLD2[9] and Rosetta[4] (in a process similar to the Pcons-Fold[3] protocol), representing one distance geometry and one fragment-based method. Since the contact information is the same across the compared methods, we are comparing the ability to generate models from a set of predicted distances or the equivalent contacts.

The CASP12 set of FM domains with public structures available was used to compare the methods. There are no structures in the DMPfold training set in the same ECOD[21] T-group as these domains. This set contains 22 domains ranging from 55 to 356 residues in length. Figure 1 shows example models generated by DMPfold. Supplementary Fig. 1 shows the DMPfold run time of the CASP12 FM domains. Table 1 shows the performance of each method on this set when different numbers of generated models are considered. We use TM-score as a measure of global similarity between a model and the native structure[22]. It can be seen that DMPfold has the best top-1 and top-5 performance of the 3 methods. In fact, DMPfold shows little variation in generated structures for a given input, and effectively produces a single output. This top-1 accuracy is very useful in practical terms, as a biologist requiring a predicted structure would prefer to work with a single model rather than 5 different possible models. Rosetta can produce models with a better mean TM-score amongst a pool of 2000, but finding this most accurate model from the rest is a difficult, and in many cases

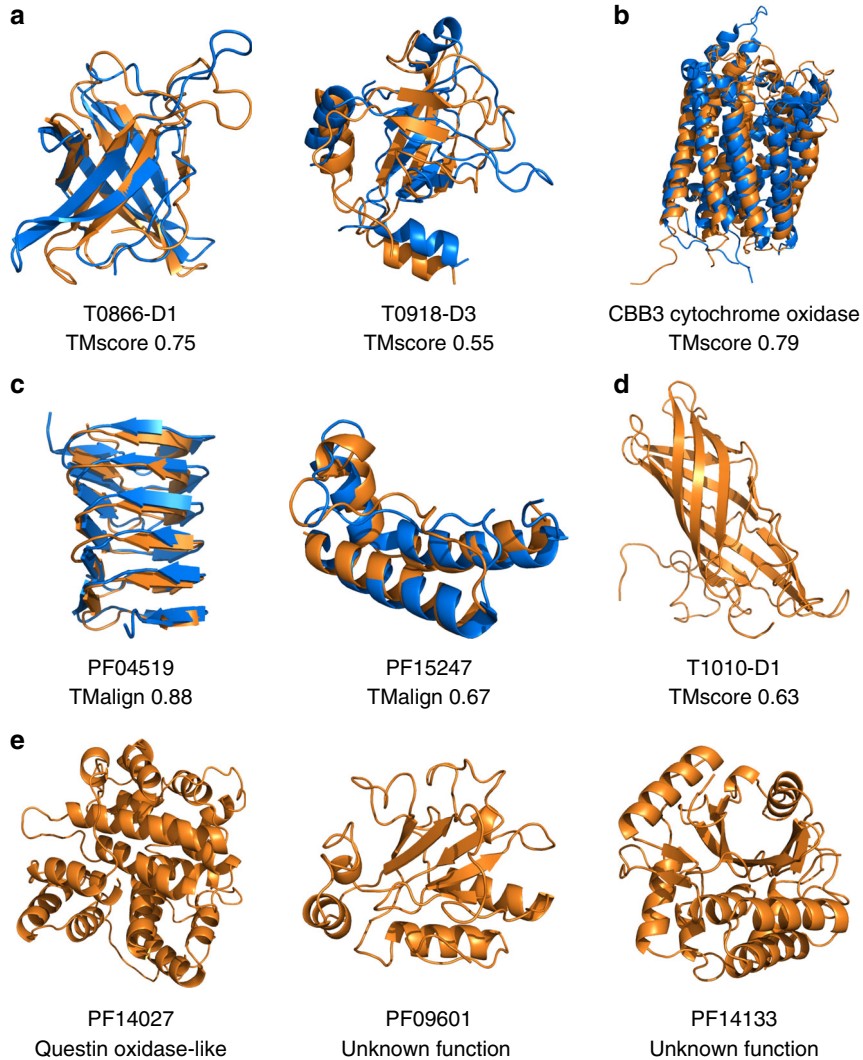

**Fig. 1** Examples of DMPfold models. In each case the model is shown in orange and the native structure, if available, is shown in blue. **a** CASP12 FM domains. **b** A membrane protein from the FILM3 set. **c** Pfam families with available structures, used as a validation set. **d** CASP13 FM target T1010-D1, where DMPfold produced the best model at CASP13 (native structure not public). **e** Models displaying novel folds for Pfam families without structures

**Table 1 TM-scores of models generated by each method on CASP12 FM domains**

| Method | Best from $n$ models | Mean TM-score | Median TM-score | Minimum TM-score | Maximum TM-score | TM-scores above 0.5 |
|---|---|---|---|---|---|---|
| DMPfold | 1 | 0.46 | 0.49 | 0.20 | 0.75 | 11/22 |
| DMPfold | 5 | 0.46 | 0.49 | 0.20 | 0.75 | 11/22 |
| CONFOLD2 | 1 | 0.38 | 0.35 | 0.16 | 0.69 | 5/22 |
| CONFOLD2 | 5 | 0.42 | 0.42 | 0.17 | 0.69 | 8/22 |
| Rosetta | 1 | 0.38 | 0.36 | 0.17 | 0.73 | 4/22 |
| Rosetta | 5 | 0.43 | 0.41 | 0.20 | 0.73 | 8/22 |
| Rosetta | 2000 | 0.50 | 0.49 | 0.25 | 0.75 | 10/22 |

In each case a number of models is generated and the highest TM-score to the native structure from the models is recorded for that domain. The mean, median, minimum and maximum are across these highest scores for the 22 CASP12 FM domains with available structures

impossible task. In addition, even when running Rosetta 2000 times, it still gives a structure with a TM-score above 0.5 for one fewer domain than DMPfold generating just a single model. DMPfold is therefore a robust and computationally efficient way to use covariation to obtain accurate 3-D protein models directly from available sequence data alone.

The distribution of TM-scores across the best of 5 models generated for each domain is shown in Fig. 2a. DMPfold generally

produces better models than CONFOLD2 and Rosetta, and is able to produce high-quality models with a TM-score of at least 0.7 for 3 domains with accurate contact predictions. The per-domain accuracy of models generated by DMPfold and CON-FOLD2, which both use CNS to generate models from distance constraints, is compared in Fig. 2b. The corresponding plot for Rosetta is shown in Fig. 2c. The benefit of iterations to DMPfold is shown by the fact that 19 of 22 domains show higher TM-score

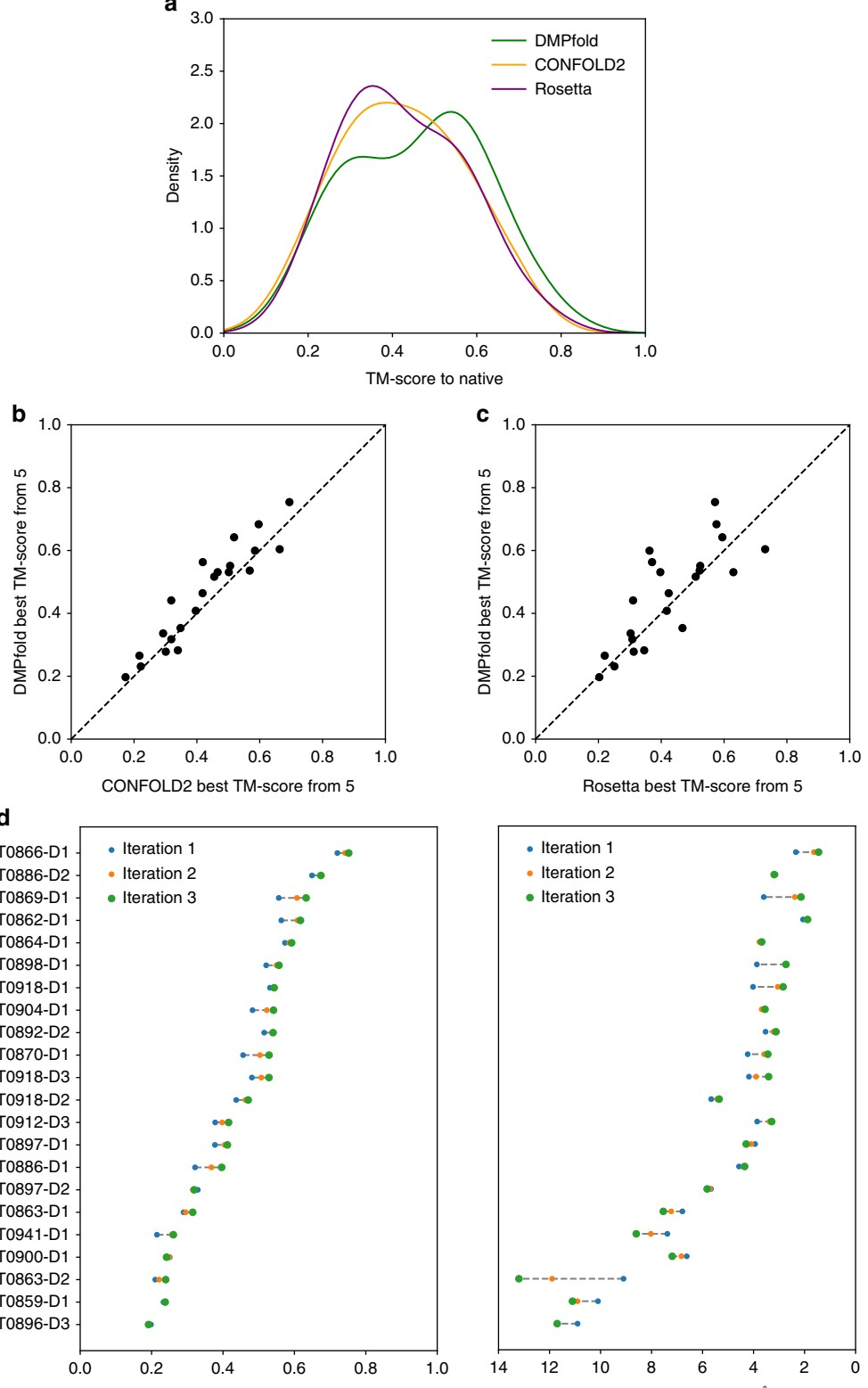

**Fig. 2** DMPfold results on CASP12 FM domains compared to existing methods. **a** Distribution of TM-scores for the best of the top 5 models for each CASP12 FM domain. **b** Comparison of DMPfold and CONFOLD2 best of top 5 models. The dashed line indicates the point of equal quality models between the two methods, which both use CNS. **c** Similar to **b** but for Rosetta. **d** The change in TM-score and absolute distance error with DMPfold iterations for each domain. Domains are ordered by decreasing iteration 3 TM-score

at the last iteration than at the first iteration, as shown in Fig. 2d. Over the course of the iterations, 3 domains move from a TM-score below 0.5 to a TM-score above 0.5. The benefits of using distance constraints rather than contact constraints were also examined. DMPfold was run with 8 Å constraints for residue pairs with a cumulative likelihood of at least 0.5 for bins up to 8 Å. Other aspects such as the iterations, torsion constraints and H-bond constraints were not changed. In this case the TM-scores have mean 0.43, median 0.43 and 10 domains have TM-score above 0.5. By comparison to Table 1, it can be seen that using distances produces better models than using contacts alone.

The importance of the three constraint types (distance, torsion and H-bond) to DMPfold is shown in Supplementary Table 1. Whilst distance constraints are required for successful structure prediction, adding torsion and H-bond constraints to the distance constraints does lead to improved performance, and the best performance is achieved when all three constraint types are combined. The prediction of hydrogen bonds using deep learning and use of these in model generation is a novel contribution of DMPfold. The hydrogen bond predictor is accurate, with a mean of 79% of predicted hydrogen bonds present according to DSSP[23] across the CASP12 FM domains. These predictions take into account the directionality of the hydrogen bonds (i.e., which residue is acting as donor and which is acting as acceptor), which further helps constrain the models.

Comparing DMPfold to the CASP12 server models indicates methodological progress in the field, and is a fair comparison as DMPfold in this case uses sequence data from the time (see the Methods section). The leading servers Zhang-Server and BAKER-ROSETTASERVER both obtained TM-scores above 0.5 for 8 of the 22 FM domains when considering the top model only, compared to 11 of 22 for DMPfold.

The accuracy of the DMPfold distance predictions can also be assessed. For the CASP12 FM domains, predictions where the bin with the maximum likelihood is less than 15 Å were compared to the true distances. The absolute error between the centre of the bin with maximum likelihood and the true distances has a mean of 5.6 Å and a median of 3.1 Å, indicating predictions good enough to build accurate models. Thirteen of 22 cases show lower mean absolute error at the last iteration than at the first iteration, as shown in Fig. 2d. The usefulness of distance bounds for modelling is highlighted by the number of bounds used that were correct, i.e., satisfied in the native structure. At the last iteration an average of 1399 out of 3640 bounds (38.4%) were satisfied per domain. This compares to an average of 125 out of 226 (55.1%) contacts that were correct, where a contact is considered present if the sum of likelihoods in bins up to 8 Å is at least 0.5. Therefore using distance bounds gives more than 10 times the number of correct structural constraints than using contacts alone for modelling.

**DMPfold on transmembrane targets**. Despite structure determination advances in recent years, transmembrane proteins (TMPs) still have very sparse structural coverage even though they play critical roles in the cell[24]. A general method for model generation should be applicable to TMPs without modification. We ran DMPfold without modification on the FILM3 dataset of TMPs[25], all of which have sizeable sequence families. There was no overlap with the DMPfold training set. Of the 28 proteins, all but 2 had a DMPfold top model with a TM-score of at least 0.5 to the native structure. The mean TM-score of these top models was 0.74, compared to 0.60 for the final refined model in the FILM3 paper results. The same sequence alignments were used for DMPfold as in the FILM3 paper, making this a fair comparison. The distribution of TM-scores and per-protein values for each

method are compared in Fig. 3. These results show that DMPfold is able to generate high quality models for TMPs without specific consideration of TMP properties.

**DMPfold on Pfam families**. A Pfam validation set was constructed consisting of 1154 Pfam families with available structures that were not related to the proteins in the DMPfold training set. These structures were either annotated in Pfam or found using HHsearch with an $E$-value threshold of 0.001; see the Methods section. DMPfold was run on target sequences from these families to return a single structure per family. The accuracy of the models is shown in Fig. 4b. Similar to the CASP12 results and other recent top methods, 52% of models have a TM-score (using TM-align; see the Methods section) of at least 0.5 to the known structural template, indicating the correct fold. The subset of families with structural templates that have an HHsearch $E$-value below 1E-6 gives 66% of models with a TM-score of at least 0.5, indicating that some incorrect predictions may be due to differences between the target sequence and the template. Interestingly, the performance of DMPfold is reasonably robust when predicting folds not seen in the training set: 46% of models in a unique ECOD X-group from the training set (i.e., which have dissimilar structure and no homology to any protein used in training) have a TM-score of at least 0.5 to the known structure. To give users a better indication of the reliability of a given model, we developed a simple neural network to predict the TM-score of a model from its sequence length, family effective sequence count and a measure of how well the final model matches the initial distance prediction distributions. This network has a precision of 82.5% under cross-validation when predicting whether the TM-score is greater than 0.5. Of the models incorrectly predicted to have a TM-score of at least 0.5, 56.3% turn out to still have a TM-score of at least 0.4, indicating that many of the false positive models may still be useful in terms of approximate chain topology. The correlation between real and predicted TM-scores is shown in Supplementary Fig. 2 and has a Pearson correlation coefficient of 0.733 under cross-validation.

We ran DMPfold on the 5214 Pfam families without an annotated structure or available template in the PDB and a target sequence length between 50 and 800 residues. The lower limit of 50 was chosen for comparison to the Baker group study[19], which used this limit. The numbers at each stage of the modelling pipeline are shown in Fig. 4a. After filtering for those predicted to have a TM-score of at least 0.5, 1475 Pfam families remained. This represents models for 25.0% of the so-called dark Pfam families[26]. Two recent studies have also made predicted structures available for Pfam families[13,19,20]. The overlap of our provided models with the Baker group study and PconsFam is shown in Fig. 4c. We provide 977 models not in either of the other sets. We predict 231 novel folds by comparing structures to the whole PDB with TM-align, and treating models as novel folds if the highest TM-score to the PDB is lower than 0.5. This criterion is the same as used in ref. [19]. This relatively low discovery rate of novel folds is similar to that observed in the various experimental structural genomics projects, e.g. ref. [27]. Some potentially novel folds and validation set structures are shown in Fig. 1.

The accuracy of models present in both our Pfam validation set and in the sets provided by the previous studies are shown in Fig. 4d, e. DMPfold gives similar quality models to the Baker group study despite not having an explicit refinement step, not using metagenomic sequences and taking considerably less compute resources. DMPfold generates better models than PconsFam in every case, indicating how the field has progressed from using predicted contacts alone for generating models to the use of richer constraints, particularly distance distributions.

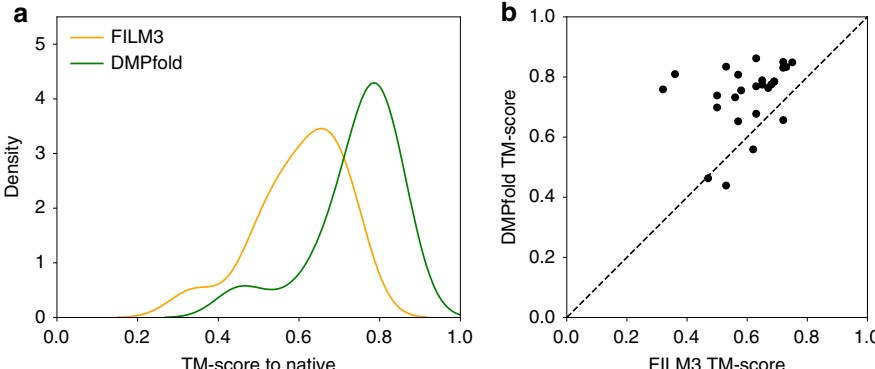

**Fig. 3** Performance of DMPfold on transmembrane proteins. **a** Distribution of TM-scores for the FILM3 TMP dataset. One model is generated for each of the 28 proteins. The FILM3 results are the final refined models from the FILM3 paper[25]. **b** The per-protein correlation of TM-scores. The dashed line indicates the point of equal quality models between the two methods

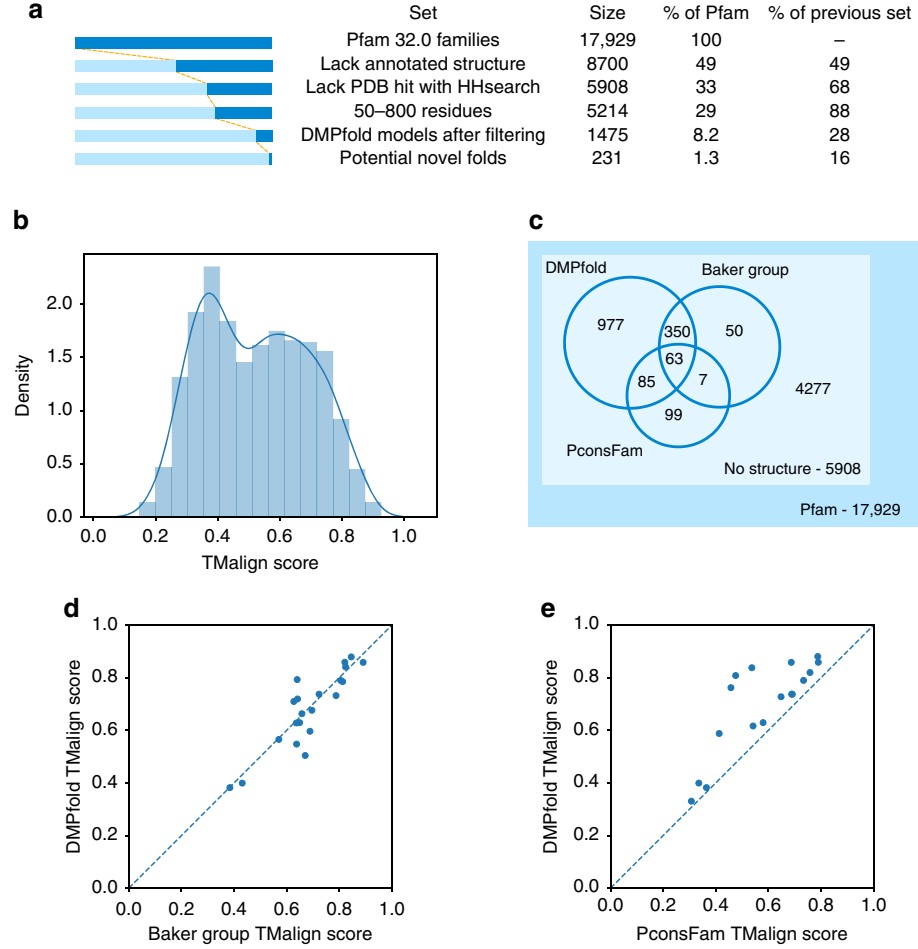

**Fig. 4** DMPfold run on Pfam families. **a** Number of Pfam families at each stage of the analysis. Each set is a subset of the previous set. **b** The TM-scores using TM-align of generated models to the native structure for the validation set of Pfam families with available structures not used for DMPfold training. **c** Overlap of high confidence models provided by DMPfold with two other studies that generated models for Pfam families. **d** Comparison of models after refinement provided by ref. [19] with our models where a native structure is available. **e** Comparison of high confidence models provided by ref. [13] with our models where a native structure is available. These are not the same families as in **d**

As shown in Fig. 5a, DMPfold is able to generate models for Pfam families with few sequences available. When the alignment contains 50–100 sequences the chance of generating a model with the correct fold is 22%. This rises to 38% for alignments with 100–200 sequences, 57% for 200–500 sequences, 58% for 500–1000 sequences, 66% for $10^3$–$10^4$ sequences and 84% for $10^4$ or more sequences. Improved success on smaller alignments is a major advantage of DMPfold that makes it applicable to many proteins previously inaccessible to tertiary structure prediction. It also indicates general development in the field

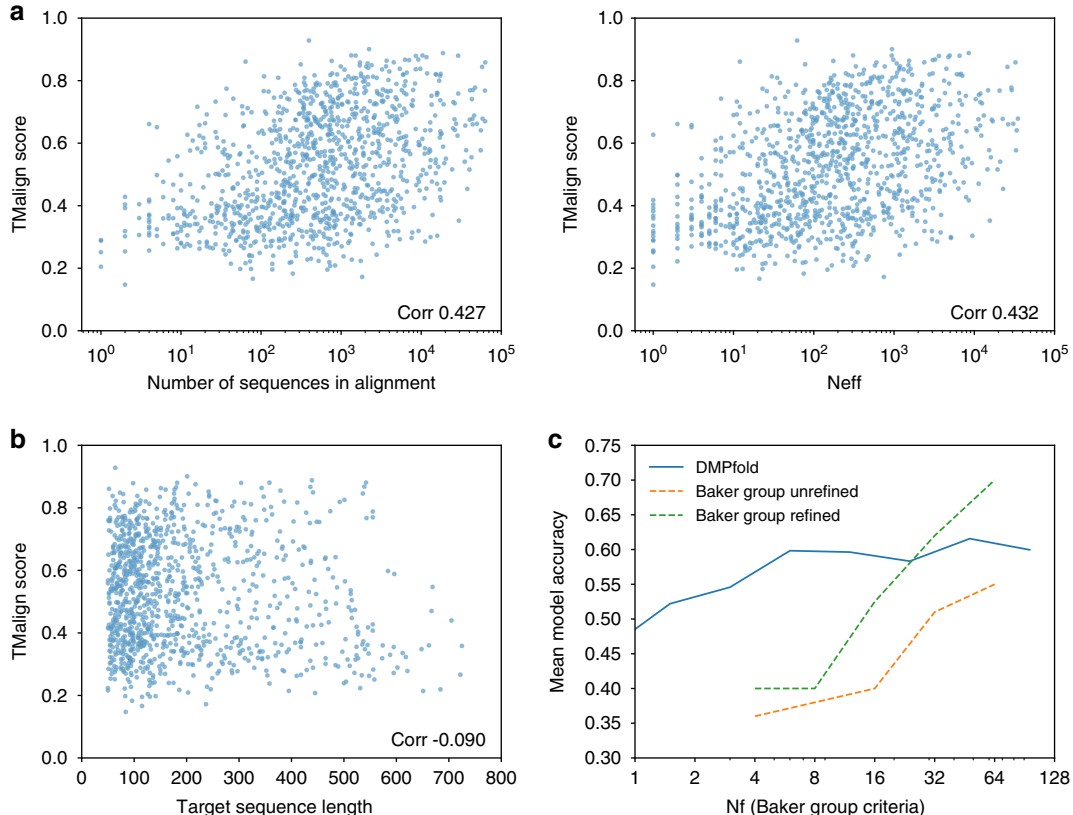

**Fig. 5** DMPfold predictions are robust to variations in MSA composition and sequence length. Evaluations are made on the Pfam validation set. **a** Correlation of TM-align score with alignment depth and effective sequence count $N_{eff}$, defined in the Methods section. The Pearson correlation coefficients are shown. DMPfold is able to generate accurate models for some Pfam families with fewer than 100 sequences in the sequence alignment. **b** There is little correlation of model accuracy with target sequence length. **c** In order to compare with ref. [19], we calculated the $N_f$ with their criteria and plotted the mean model accuracy of models in bins of $N_f$ values. $N_f$ values were calculated as described in ref. [19], where an 80% identity threshold was used for clustering. Values were read off the graph in Fig. 2 of ref. [19] and added here. It is important to note that the proteins used to obtain our values were different to theirs. It can be seen that DMPfold is effective at lower effective sequence counts

from approaches such as PSICOV[28] and CCMpred[29], which require large alignments to work. This shows the advantage of using deep learning approaches that can make use of sparse data and use a trained knowledge of proteins to fill in the gaps and make more accurate predictions.

Comparing to the Baker group study further shows the ability of DMPfold to work with shallower alignments. Model accuracy at different alignment sizes, represented by an effective sequence count $N_f$, is shown in Fig. 5c. It can be seen that DMPfold has a relatively flat line, and is able to generate accurate models even when relatively few sequences are available. Also shown superimposed on the plot is a similar plot from the Baker group study. Although these two lines are not directly comparable because different proteins were used in each set, they are indicative of DMPfold being able to give good performance even with shallower sequence alignments. For deeper alignments, DMPfold is able to give unrefined models of a quality close to the refined models from the earlier study.

There is little correlation of model accuracy with sequence length, as shown in Fig. 5b. The prediction of residue pair distances and iterative improvement of models allows domains up to around 600 residues in length to be modelled accurately. Beyond this, the accuracy falls with the default parameters used. This dropoff in accuracy may also stem from the DMP training set, which had a maximum chain length of 500 residues. In addition, some of these longer proteins have multiple structural domains despite being from one Pfam family, which can make

modelling and assessment hard. Mirror topology effects can also be an issue—see the Discussion. We recommend that users treat DMPfold models for proteins of more than 500 residues with caution and consider splitting them up. The benefit of iterations to DMPfold is shown in Fig. 6. In the example shown, a loop moves from an initially incorrect predicted position to a more native-like position after 2 iterative predictions. One future use of DMPfold could be as a refinement tool for structures that are broadly correct.

The compute time required to generate a model from a 200 residue sequence is about 3 h on a standard single-core desktop computer (see Supplementary Fig. 1), including the time required to build the multiple sequence alignments (MSAs) from the target sequence. Model generation takes about 1.5 h of this time. This means that any researcher can generate 3-D models for proteins of interest, as the DMPfold software and data is freely available. Also, this efficiency means that we will be easily able to maintain a continuously updated set of de novo modelled Pfam domains, which will be distributed via the Genome3D resource[30].

**Newly modellable regions in model proteomes.** Limiting ourselves to high-confidence predictions in dark Pfam families, we evaluated the extent to which our predictions extend the structural coverage of the proteomes for *Homo sapiens* and 13 other model organisms. The left-hand pie chart in Fig. 7 shows the number of residues in UniProt[31] entries for each taxon for which DMPfold can provide confident models. The values for this pie

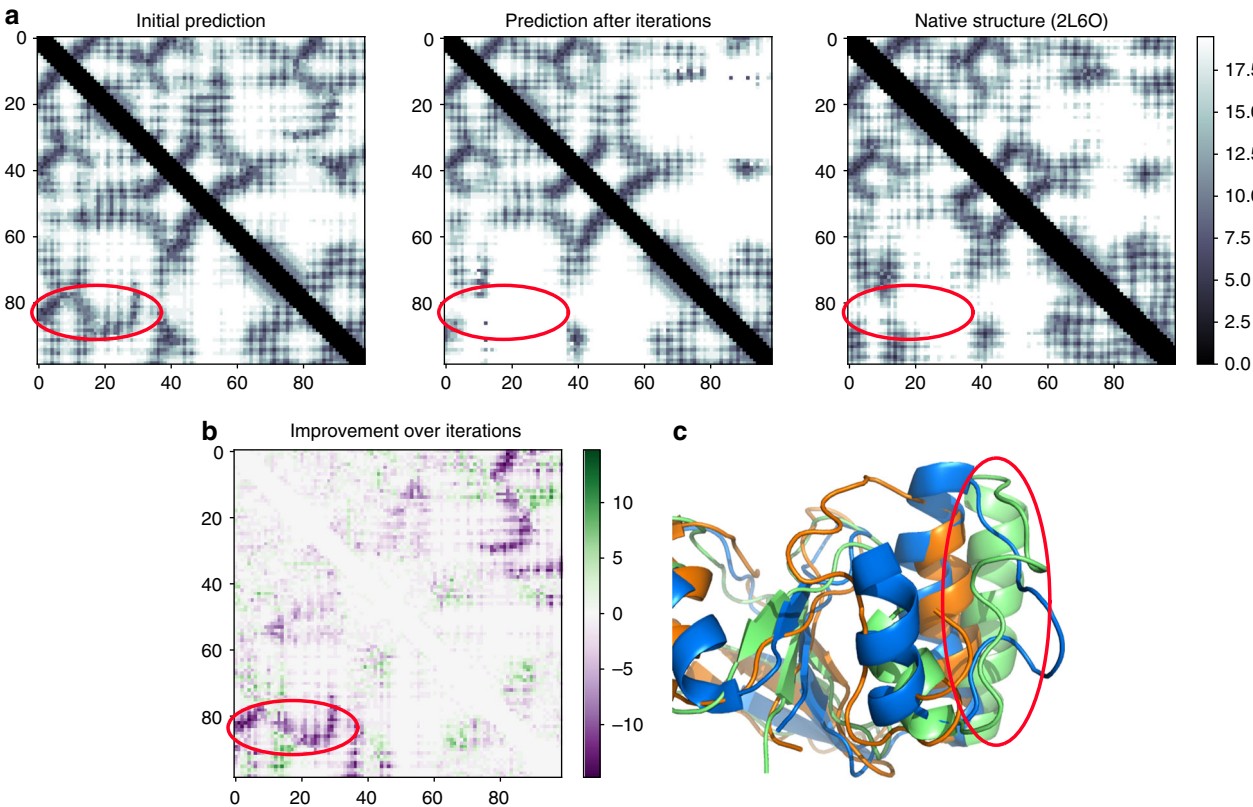

**Fig. 6** An example of model accuracy increasing after iterations. The model is for Pfam family PF13642. **a** Distance maps of the initial prediction, the prediction after iterations and the native structure. In each case the value at $i, j$ is the centre of the distance bin with the maximum likelihood between residues $i$ and $j$. **b** The change of the absolute error in distance from the initial prediction to the prediction after iterations. A negative value indicates an improvement with the iterations. **c** The improvement is shown on the structure. The native structure (PDB ID 2L6O) is in blue, the initial model is in orange and the model after iterations is in green. The loop region indicated in red throughout, and the following helix, are closer to the native structure in the prediction after iterations than the initial prediction

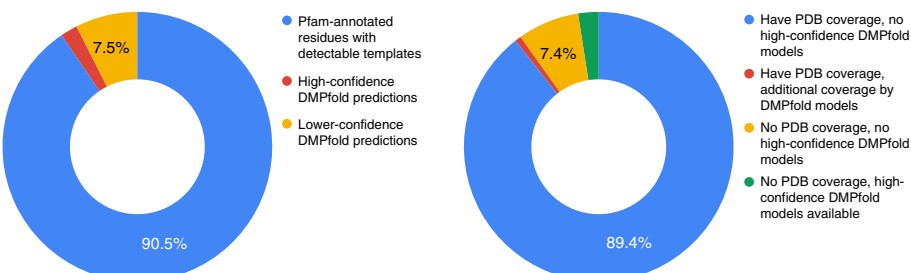

**Fig. 7** Coverage of proteomes by DMPfold models. Left: Pie chart showing the fraction of Pfam-annotated amino acid residues in a number of proteomes for which templates or PDB matches are available (blue). Of the remaining residues, the fractions covered by high-confidence DMPfold models (red) and lower-confidence models (orange) are marked. Right: Pie chart of UniProt entries in several proteomes. Entries are first split into whether or not they are (at least partially) covered by PDB matches or templates. Within each split, we then assess the fraction of entries that either have or do not have high-confidence DMPfold models available. The green fraction indicates entries for which de novo models provide the only structural information currently available. Data for each fraction of each pie chart are summed over several proteomes (full data in Supplementary Tables 2 and 3)

chart were summed over the various proteomes considered; full data appear in Supplementary Table 2. In terms of the total numbers of amino acid residues covered by DMPfold predictions for the taxa in Supplementary Table 2, we find that our predictions lead to a modest fractional increase in coverage of at most 4.3% and typically no more than 2% of existing structural coverage (depending on the specific proteome considered). This reflects the fact that these dark families are significantly less abundant (on average) than families with existing structural coverage in the individual proteomes considered (assessed by

one-sided Wilcoxon rank-sum test at $\alpha = 0.05$), which may be a bias that arises from structure determination efforts, especially the structural genomics initiatives, focussing more attention on larger sequence families, especially those that repeat frequently across model organism proteomes[32]. For additional perspective, we also calculated the number of UniProt entries for each taxon for which DMPfold produces a new model, and the number of entries out of these for which no other PDB hits or templates can be detected. A summary of this data appears in the right-hand pie chart of Fig. 7 and the raw data appear in Supplementary Table 3. We find that

across the proteomes considered, most entries are already covered at least partially by PDB structures, DMPfold typically provides new annotations for thousands of UniProt entries across the various proteomes, and the majority of these models cover entries that are not covered (even partially) by PDB structures. 2.5% of all the UniProt entries considered get their first structural annotations from high-confidence DMPfold models (green fraction of right-hand pie chart in Fig. 7).

## Discussion

In the last few years, contacts predicted from covariation data have become accurate enough to produce good quality protein models. However, contact prediction and model generation have generally been treated as separate steps. DMPfold combines the two stages in an iterative process, allowing constraints to be refined based on the models produced. Prediction of distances rather than binary contacts provides considerably more information for model building. Modifications such as these will be necessary for template-free modelling to move from identifying the correct fold to generating high-quality models useful for studies such as ligand binding. Nevertheless, methods that can generate models for thousands of target sequences in days on a standard research cluster are going to become increasingly important as we move further into the modern genomic era.

As expected, more accurate contact predictions (and, by extension, more accurate distance predictions) give more accurate models. Of the 12 CASP12 FM domains with top $L$ predicted contact accuracy of at least 0.5, 8 have a TM-score of at least 0.5 to the native structure. As the volume of available sequence data increases, the corresponding increase in distance information should lead to more accurate DMPfold models.

In the recent CASP13 experiment, a prototype version of DMPfold was ready in time to make predictions for about two thirds of the prediction targets, and the current version used for the last third. Results from the CASP website indicate that our group, Jones-UCL, submitted models with the correct fold (TM-align score at least 0.5) for 24 of 43 (56%) FM domains. Results for other leading methods using deep learning to predict distances included 31/43 (72%) for A7D (AlphaFold from DeepMind) and 24/43 (56%) for RaptorX-DeepModeller[14]. For comparison, the leading group in CASP12 achieved 22/56 (39%) of predictions with the correct fold. In CASP13, a number of groups employing deep learning techniques, including us, moved this fraction up to around 60%. Whilst our CASP13 results were not quite at the level of the best-ranked method, AlphaFold, DMPfold gives competitive results with a much shorter computation time[33] and is freely available software. DMPfold makes the recent advances in protein structure prediction available to the community to run at genome-scale.

In principle, DMPfold is also applicable to multi-domain proteins in its current form. However, in testing we noticed that in some cases, individual domains can be predicted as topological mirror images[12]. These are distinct from simple mirror images in that topological mirror images have the correct stereochemistry and handedness of alpha-helices (as a result of CNS structure calculations), but differ in the overall topology of the protein chain. In addition, in our predictions they also have seemingly well-packed hydrophobic cores, making it non-trivial to distinguish a mirror topology from the correct fold. During testing, we observed that for predictions on some two-domain proteins, DMPfold often produces structures in which one domain is in the correct fold, while the other is a topological mirror. A rudimentary experiment using distance data sampled from native structures suggests that this problem arises due to an insufficient number of inter-domain distance constraints being supplied to

the CNS calculation. We expect that further improvements in the accuracy of the distance predictions in DMPfold, as well as the method by which distance constraints are assigned, will alleviate this problem.

As we stated earlier, the models provided for the 1475 Pfam families had an 82.5% likelihood of having a correct fold. It is also noteworthy that there will be a number of correct models for the remainder of the modelled 5214 dark Pfam families that did not pass the filtering criteria. Assuming the TM-score prediction results on the validation set are comparable, we can estimate that 18% of the rejected 3739 families, equalling around 670 families, will have models with the correct fold. Considering additional information, such as conserved functional sites, could help identify some of these cases. By combining this with the filtered models we predict the overall number of correct folds we predict for dark families to be 1887. The continual increase in the number of available sequences will also make more families amenable to modelling in the next few years. Based on the change from Pfam 29.0 to 32.0, there is an increase of around 40% per year in the number of Pfam sequences. In one year this would move around 273 of the 3739 (7%) Pfam dark families without reliable DMPfold models from fewer than 50 sequences in the alignment to more than 50 sequences. This change should make these families more likely to be modelled accurately in a year's time (see Fig. 5). While this paper was under review, 9 of the Pfam families modelled at high confidence had a structure released in PDB for the first time. 8 of these 9 models have a TM-align score of at least 0.5 to the deposited structure, with a mean TM-align score of 0.62; see Supplementary Table 4. This set acts as a further validation set for DMPfold, and the fraction of correct folds matches the precision of the model accuracy predictor. The one model with TM-align score less than 0.5 corresponded to a de novo-modelled region of a 4 Å resolution cryo-EM structure, which may not be reliable.

One important difference between this study and the earlier study from the Baker group is that we decided against including metagenomic sequences in our alignments. There were two main reasons for this. Firstly, and most importantly, we wanted to emphasise the methodological improvement afforded by the use of deep learning in de novo structure prediction here rather than simply the growth in sequence data since the earlier study was performed. Secondly, we felt that the difficulty in tracking the provenance of metagenome data (e.g., the lack of reliable source organism information in many cases) makes the choice of using it in the public annotation of genome data rather ambivalent. Sticking to sequences taken directly from the well-curated UniProt data bank[31] makes it far easier to maintain the annotations and to use phylogeny, for example, to link structure to function downstream of the modelling process. Obviously on a case by case basis, or where the accuracy and coverage of 3-D modelling are the sole factors to consider (in the CASP experiment for example), then the sensible approach would be to include as many homologous sequences as possible from any available source.

An obvious limitation of the current study is that it only explores the de novo modelling potential for the parts of proteomes which match Pfam domains. Considering an average 'proteome' as the sum of all UniProt entries across the proteomes listed in Supplementary Tables 2 and 3, on average just under half of the entries have at least one Pfam annotation. For the human proteome this is only 27.9% of entries, however, although in terms of amino acid residues, just over half (50.4%) is covered by Pfam annotations.

How much of the remaining half of the human genome could still be correctly modelled by DMPfold is hard to assess. The remainder will, of course, include a mixture of disordered or

unstructured proteins, coiled-coil regions and other general low-complexity features. Some of it, however, will be modellable regions which Pfam simply has not reached due to a lack of homologous sequences in standard databases. Even the disordered domains could have a viable native structure under the right conditions e.g., through formation of multimers or binding to cognate ligands. For future work, it would be interesting to scan the proteome regions where no Pfam domains can be found and simply look for high-complexity sequences which produce reasonably deep alignments with HHblits. DMPfold could, in those cases, provide models which could help identify these regions as potential new superfamilies or help identify them as distant members of existing families through structure-based comparison. A further application of DMPfold could also be to help provide models which could distinguish true protein-coding genes from pseudogenes or mistranslated regions.

Although the fact that we are able to confidently annotate 25% (1475/5908) of the dark Pfam domains using DMPfold shows the power of deep learning and sequence covariation assisted de novo modelling, the additional coverage of the proteomes, perhaps at first sight, looks surprisingly low. Only an extra 1.56% of the human proteome by residue is covered by domains with new structures determined by DMPfold. This is not that surprising, however, as the largest families, which might often appear in multiple tandem repeats in a single protein, will have been prioritised for experimental structure determination in the past. The fact that we have been able to structurally annotate a further 790 human proteins with high confidence (or 8525 proteins across all 14 model organism genomes), where no prior structural information was known at all, is the key result. These very dark proteins could be keys to new biology yet to be discovered in the genomes, particularly those domains which are predicted to have entirely novel folds.

## Methods

**Deep learning components.** An overview of the DMPfold pipeline is shown in Fig. 8. DMPfold uses a set of deep neural networks similar in architecture and methods of training to our DeepMetaPSICOV (DMP) contact prediction method, which is described separately[17]. However, in this case the neural networks are used to predict inter-residue distance probability distributions (between Cβ atoms or Cα atoms for glycine), main chain hydrogen bond (H-bond) donor/acceptor pairs and torsion angles rather than just contact maps. DMP and DMPfold use exactly the same input features as shown in Supplementary Table 5, and are meta-approaches that combine different sources of covariation data from alignments using a deep residual neural network to predict contacts, along with the raw residue-residue covariance matrix as employed in the DeepCov contact prediction method[34].

For predicting main chain H-bonds, the neural network architecture is exactly the same as that of DMP itself (Fig. 9b). The only difference is that the output map represents H-bond donor (rows) and acceptor (columns) contacts, where the donor (N) is within 3.5 Å of the acceptor (O). Unlike a normal contact map, therefore, the H-bond map is not expected to be symmetric.

The DMPfold distance predictor differs slightly from the architecture of DMP. The key difference is that instead of predicting binary contacts between residue pairs, the DMPfold distance predictor outputs a probability distribution in the form of a distance histogram for each residue pair. This is achieved by the use of a softmax output layer with 20 output channels, with each channel corresponding to the likelihood of each residue pair being between two predefined distances. The distance bins used for the output channels are 3.5–4.5 Å; 7 bins running from 4.5 Å to 8 Å in steps of 0.5 Å; 11 bins running from 8 Å to 19 Å in steps of 1 Å; and a final bin for all distances 19 Å or greater. As the underlying distance matrix must be symmetric, symmetry of the final output tensor ($O$) is enforced (for inference only) as follows:

$$O_{\text{final}} = \text{Softmax}(O + O^{\text{T}}) \qquad (1)$$

This symmetry enforcement also serves to ensemble the independent upper and lower triangle prediction outputs of the DMPfold network.

The model architecture is shown in Fig. 9a. Predicting distance distributions is clearly a more complex problem than simply predicting binary contacts, and so to increase the representation power of the DMP network architecture, rather than increasing the number of layers, which would have required too much GPU memory during training, we replaced the usual two convolutional layers in each residual block with a single convolutional maxout layer[35], with 4 hidden maxout

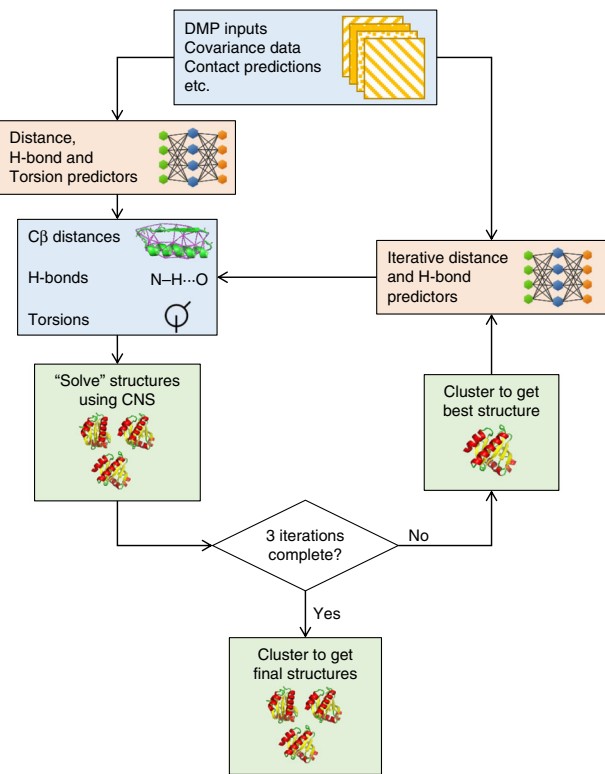

**Fig. 8** Overview of the DMPfold pipeline. Initially inter-residue Cβ distances, H-bonds and torsion angles are predicted from DMP inputs. These are used to generate models with CNS, and a single model is used as additional input to refine the distances and H-bonds. After 3 iterations a final set of models is returned

units per layer. Rather than relying on a separate nonlinearity e.g., the ReLU activation functions used in DMP, a maxout unit takes the maximum feature across multiple affine feature maps to produce a learned intrinsic nonlinearity. A single maxout network layer with more than two hidden maxout units works on its own as an efficient universal approximator of any continuous function, and so the ability of multiple convolutional layers to approximate arbitrary continuous functions can be reproduced by just a single maxout layer. The maxout layer used for initial dimensionality reduction (from 501 channels to 64) is the same as the one used in DMP itself and thus has 3 hidden maxout units rather than the 4 used in the residual blocks. The dilation rate $d$ for each residual block is shown in Supplementary Table 6. As halving the number of convolutions per block reduces the maximum receptive field size, an additional dilation of 128 was added to the maxout network to compensate.

Training of all models was performed using the Adam optimiser[36] for 75 epochs with default parameters ($\beta_1 = 0.9$, $\beta_2 = 0.999$, maximum learning rate of $10^{-3}$), the final model weights were those that produced the lowest cross-entropy loss on the validation set (5% of the training cases held out). Minibatches of size one were used for forward and backward passes due to GPU memory limitations, although gradients were accumulated and averaged across minibatches of size 8. All other aspects of training, including data augmentation procedures were out as previously described for DMP[17]. The training set here was based on the same 6729 protein chains, ≤500 residues in length, with non-redundancy at the 25% sequence identity level and no unresolved main chain atoms. A 5% subset was kept aside for validation rather than training, and any chains corresponding to CASP11 targets were also excluded from training for validation of the 3-D modelling procedures (see below). A further 9 chains were excluded from training as they overlapped with proteins in the FILM3 test set. No additional cross-validation was required for the CASP12 test set, as the data set was assembled from data available prior to the start of the CASP12 experiment, including the HHblits[37] HMM library (uniprot20 2016_02 release).

**Model generation using CNS.** CNS[11] is used to generate models from pseudo-NOE information derived from the DMP distance distributions, H-bond maps and torsion angles. In the first iteration of DMPfold, the contact maps and H-bonding maps (asymmetric donor-acceptor pair maps) are predicted. Upper and lower distance bounds, and predicted H-bonds are converted into NOE-like constraints. For the Cβ-Cβ distances (Cα atoms for glycines), the bounds for the maximum likelihood bin is taken as the starting point. These bounds are then grown

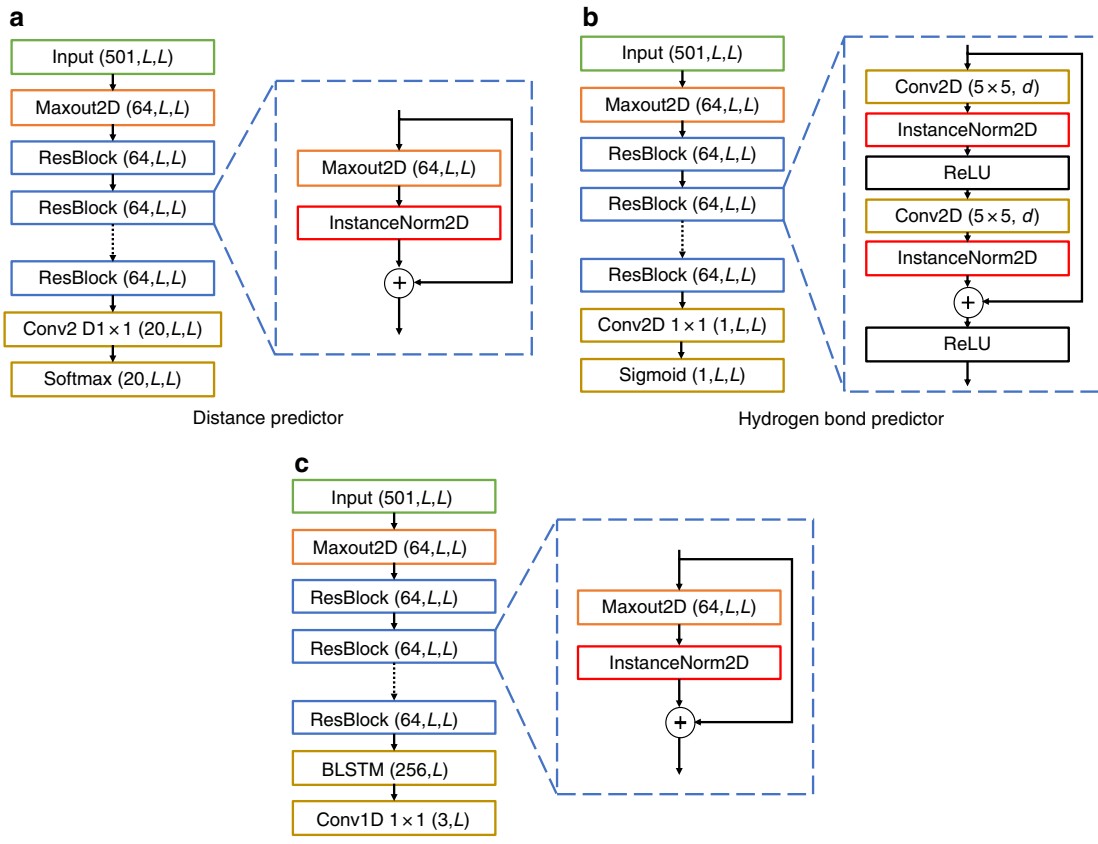

**Fig. 9** DMPfold model architectures. DMPfold uses three predictors, all of which are deep, fully convolutional residual networks. Each uses a total of 18 residual blocks, comprising convolutional layers with a mixture of standard and dilated 5 × 5 filters. Where numbers are included in parentheses, these are the dimensions of the tensor output by the respective layer. For the iterative versions of the distance and H-bond predictors, the input tensor includes an extra feature channel composed of values taken from structures in the prior iteration (for a total of 502 channels). See the Methods section for full details

accretively to encompass neighbouring bins in order of likelihood until the total likelihood reaches a set threshold. A total likelihood threshold of 0.4 was found to be optimal, though the overall method is relatively insensitive to changes in this threshold. This method of selecting bounds means that less confident predictions result in wider bounds. Bounds are not generated for cases where the maximum likelihood bin is the unbounded last bin, and the last bin is also excluded from the likelihood accretion procedure. This means that all upper bounds provided to CNS are <19 Å. For the binary output H-bond prediction network, a binary likelihood threshold of 0.85 was used to decide whether to consider the predicted H-bond or not. Again, although 0.85 was found to be slightly optimal, other threshold values over 0.4 perform almost equally well. The main chain H..O and N..O distance constraints input to CNS for the predicted H-bonds are fixed according to the values observed in highly resolved crystal structures.

**Additional constraint types and iterative predictions**. In addition to the distance-based constraints, DMPfold also generates dihedral angle constraints from predicted main chain torsion angles. Torsion angle predictions are generated with the same deep residual maxout network, but with the 20-dimensional softmax output layer replaced by a bidirectional recurrent LSTM layer[38] with 128 hidden units (BLSTM in Fig. 9c), which embeds each row of the final 2-D 64-channel feature map in a single 256-D vector (concatenation of 128-D final timestep output states of the forward and reverse direction LSTM passes). As each row ends up embedded in a single vector, the LSTM layer thus transforms the 3-D tensor (64 features × L × L) into a 2-D tensor (256 features × L), which is finally reduced in dimensions to three final feature channels (3 × L), interpreted as φ, ψ, and ω angles in radians for each residue. The mean squared errors of the sines and cosines of these angles was used as the loss function. The accuracy of torsion angle prediction was observed to be comparable with the current state-of-the-art[39,40], with an average MAE (mean absolute error) of 18.8° for φ and 26.7° for ψ on a benchmark set of CASP11 FM and TBM-hard targets. As the network did not demonstrate any ability to predict rare cis-peptide conformations (ω ≈ 0°), ω angle predictions were not used as modelling constraints. Accuracy estimates of predicted φ, ψ, and ω angles were produced by training a second network with identical architecture to predict the errors of angle predictions for the training set i.e., to predict the

reliability of predictions from the first network. The loss function in this case is simply the mean squared absolute error in the predicted torsion angles. These error estimates are then used to populate the deviation fields of the CNS dihedral angle constraints file. As we saw no real benefit from calculating the dihedral constraints iteratively, for simplicity we calculate just a single set of dihedral constraints in the first iteration and used them throughout the subsequent modelling cycles.

Using the initial input constraints, a predefined number of models are generated and clustered by structural similarity. The representative structure of the largest cluster is taken, selected by estimated model accuracy using a combination of MODCHECK[41] Cβ potentials of mean force and all atom MODELLER[42] DOPE-HR scores, and this model used to seed the next iteration. The same distance and H-bond procedures described above are used, but with an additional input feature channel added, namely the Cβ-Cβ distance matrix calculated from the seed structure. This allows new distances and H-bonds to be predicted using prior information of likely Cβ-Cβ distances from the previous iteration of 3-D modelling. In this way, the combined contact prediction and structure generation procedure can evolve a better prediction at each iteration. To train these iterated models, a standard unseeded DMPfold run was carried out for each training set protein, generating 6729 ensembles of 20 models per target. At each epoch of training the iterated neural network models, structures were selected at random from the ensembles and the calculated distance matrices used to populate the extra feature channel. In this study we only trained one set of iterated models, but in the future, a small amount of improvement might be achievable by training further iterated neural network models, where DMPfold is re-run after each iteration of training to produce seed inputs for the next iteration.

Typically fewer than 5 iterations are needed for convergence—we use 3 iterations throughout the work reported in this paper. A set of 30 FM-category domains from CASP11, which had no protein domains in the same ECOD H-group level[21] as the DMPfold training set, was used to select optimum parameters (i.e., the likelihood thresholds used to derive upper and lower distance bounds) for the DMPfold model generation steps.

**Enforcement of non-overlap between training and test sets**. In order to assess the generalisation ability of any trained deep learning model, it is crucial to ensure

that the set of examples used to train the model does not share any obvious overlap with examples used to test the model. For proteins, a criterion based on sequence identity between any training and test example is commonly used. However, this is generally insufficient to rule out an evolutionary relationship or structural similarity, as many related proteins share less than 20% sequence identity. It is much more preferable to use a structural classification database such as CATH, SCOPe or ECOD. We use the evolutionary classification of domains (ECOD) database[21] to define our train-test split for all benchmarks in this paper.

**Calculation of effective sequence counts**. Effective sequence counts ($N_{eff}$) were determined as per ref. [34]. Briefly, sequences in each MSA were clustered using CD-HIT[43] at a sequence identity threshold of 62%. The number of clusters returned by CD-HIT was taken as the $N_{eff}$. When comparing results against the Baker group study[19], we used the same calculation of effective sequence count as in that work, which we denote $N_f$.

**Running DMPfold on CASP12 targets**. All of the input alignments were generated using HHblits[37] with the Feb 2016 release of the uniprot20 HMM library[31]. This ensured that only sequences available at the start of the CASP12 experiment were considered. To compare to contact-based methods, DMPfold distance predictions were converted to predicted contacts by summing up likelihoods for distance bins below 8 Å for a given residue pair and sorting residue pairs by the resulting sum. We find that these contact predictions closely match the accuracy of using DMP to predict contacts. CONFOLD2 was run with default parameters and secondary structure predictions from PSIPRED[44]. Rosetta was run similarly to the approach in the PconsFold protocol[3]. Fragment generation was carried out with default parameters and the non-redundant sequence database. Since the database of structures used for fragments was assembled before CASP12 took place, there are no homologous proteins to the CASP12 FM domains in this set. The Rosetta *AbinitoRelax* protocol was run with the "-abinitio:increase_cycles 10" and "use_-filters" options. The top $L$ predicted contacts were used as constraints where $L$ is the sequence length.

**Running DMPfold on transmembrane targets**. The sequence alignments used to generate DMPfold models for the FILM3 dataset were identical to the sequence alignments from the FILM3 paper[25], allowing direct comparison to the results presented there. None of the targets in this set are homologous to any training example, as assessed by ECOD database classification at the T-group level.

**Running DMPfold on Pfam families**. Pfam 32.0 (September 2018 release) was used[18]. The set of dark families available for modelling was taken to be the set of 8700 lacking an annotated structure minus those families with likely templates not annotated in Pfam—these were found by running HHsearch[45] of the Pfam seed HMM against the standard HHsearch PDB70 HMM library and taking hits with an $E$-value threshold of 1.0. The number of families remaining was 5908. A representative target sequence was found for each family using hmmsearch[46] to search the UniRef90 database with the Pfam HMM and taking the closest subsequence match by $E$-value. The 5214 families with target sequence lengths between 50 and 800 were taken forward for de novo modelling. Alignments were generated for each target sequence using HHblits[37] searches against the 2018_08 version of the UniClust30 database. DMPfold was run from these alignments and the top model for each family was taken forward for further analysis. Potential novel folds were determined as those with a maximum TM-score using TM-align (normalised by the model length) of 0.5 or less to any structure in the PDB.

The Pfam validation set consisted of families with available structures that were not used for the training of DMPfold. Highly likely templates not annotated in Pfam were found by running HHsearch of sequences in the PDB against the Pfam HMM and taking hits with an $E$-value threshold of 0.001. Structures were determined as not being used for DMPfold training if they were in a different ECOD[21] T-group to all structures in the DMPfold training set. The validation set consisted of 1154 Pfam families once target sequences outside the 50–800 residue range had been excluded. For each family, a single PDB structure was chosen for comparison to the model. In the case of the Pfam structural annotations, this is the structure with the highest alignment score to the target sequence. In the case of the HHsearch PDB hits, this is the closest hit by $E$-value. TM-scores were calculated using TM-align[47] rather than TM-score as the target sequence and PDB sequence are often very different and so not easily aligned by sequence. The maximum of the TM-scores normalised by the model or the PDB chain was taken, as it is possible that each could fail to cover the whole length of the other.

**Estimating model accuracy**. To produce a useful estimate of model accuracy we trained a small fully connected neural network to predict the likely TM-score for a given model. The inputs to this model were the target sequence length, the effective number of sequences in the alignment, the sum of distance histogram likelihoods, and the average distance histogram likelihood. The histogram likelihoods were derived from the softmax outputs of the first iteration distance histogram prediction where the likelihoods of the bins selected by each pairwise distance in the model were summed (or averaged), standardised in the usual way using the individual feature means and standard deviations. The EMA neural network

comprised the 4 feature inputs, two fully connected hidden layers of 10 units per layer, with 10 softmax outputs corresponding to the TM-score ranges ($0 \leq s < 0.1$, …, $0.9 < s \leq 1.0$). A SELU activation function[48] was used for the hidden layers and a cross-entropy loss function used for training with the Adam optimisation algorithm[36] and a maximum learning rate of $10^{-3}$. An expected TM-score was calculated for a model by calculating an average of each output bin ranges (midpoint TM-score) weighted by the softmax output for the bin.

To estimate the accuracy of this method for discriminating models with a correct fold, the network was trained 100 times on the Pfam validation set of 3-D models with random training/validation/test splits of the data each time. Using a predicted TM-score threshold of 0.5, we found that the validation set models with correct folds (observed TM-score > 0.5) could be recognised with a mean precision of 82.5% and a recall of 82.2%.

**Estimating structural coverage in proteomes**. We used the proteome mapping files from Pfam to assess the additional coverage provided by our predictions. After resolving secondary UniProt accessions and removing deleted entries, we counted the number of residues annotated with Pfam IDs and assess what number of residues (a) could already be annotated with 3-D structures by homology; (b) can now be confidently modelled using DMPfold; or (c) cannot be modelled either by homology or using high-confidence DMPfold models. For (b) we limited ourselves to dark Pfam families (see section "Running DMPfold on Pfam families") that could be modelled with high confidence. Pfam annotations on each UniProt entry were demarcated by the envelope coordinates provided by the Pfam assignments for the given proteome. We also assessed the number of whole UniProt entries covered at least partially by PDB entries or templates, and the number that have no structural coverage at all. We then assessed the number of entries in each category that were (at least partially) covered by high-confidence DMPfold predictions for dark Pfams. These analyses were carried out for a number of proteomes, including *Homo sapiens* and several model organisms.

**Reporting summary**. Further information on research design is available in the Nature Research Reporting Summary linked to this article.

## Data availability

The trained neural network models and Pfam 3-D models are available at https://github.com/psipred/DMPfold. All other relevant data are available from the authors upon reasonable request.

## Code availability

The deep learning components of DMPfold are implemented in PyTorch[49]. The source code and documentation are available at https://github.com/psipred/DMPfold.

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

## Acknowledgements

We thank members of the group for valuable discussions and comments. This work was supported by the European Research Council Advanced Grant "ProCovar" (project ID 695558). This work was supported by the Francis Crick Institute which receives its core funding from Cancer Research UK (FC001002), the UK Medical Research Council (FC001002), and the Wellcome Trust (FC001002).

## Author contributions

D.T.J. conceived the research, carried out the machine learning experiments and developed DMPfold. S.M.K. and J.G.G. carried out the benchmarking and genome annotation experiments. All authors contributed to the writing of the paper.

## Additional information

**Competing interests:** The authors declare no competing interests.

