## [Peer Review File · Nature Communications]

Reviewers' comments:

Reviewer #1 (Remarks to the Author):

This manuscript proposed to model protein structures using neural-network based residue distance prediction. It is a straightforward extension of the DeepMetaPSICOV (DMP) program, developed by the same group, but with contact map replaced by distance map. Similar work was done by Xu as cited in Ref [14]. There is a modest novelty in which the iterations are introduced between distance training and model generations. The work catches up the hot topic of distance-guided structure prediction, which started to become popular since the work of the AlphaFold team in CASP13. But there are substantial problems in the way that the authors present and benchmark their method. My major concern on this work is that it does not represent the state of the work.

Major concerns:

1. The method was only compared to Confold2 and Rosetta based on a small set of 22 CASP12 FM domains. First of all, Confold2 and Rosetta in the comparison do not use distance constraints. The authors converted their distance prediction into contacts and then used the converted contacts to guide Confold2 and Rosetta. Thus, it is not clear how much information was lost in the conversion and how accurate the converted contacts are. Why do not you simply use DMP contact map? Second, since you are testing your method on the CASP12 domains, why do not you directly compare your models with the released models by the CASP12 servers? This will help examine how far your method is compared to the state of the art (3 years ago) Third, the time is now post CASP13 and methods in CASP13 had a significant progress compared to the approaches in CASP12. To benchmark with the real state of the art, the method must be compared to the CASP13 servers. In fact, the Jones-UCL group performed quite poor, compared to the best CASP13 server predictors (ranked as No 23 at the list of http://predictioncenter.org/casp13/zscores_final.cgi). This raises some concern on the performance of this presented method. Finally, the 22 FM domains are really small to draw a solid conclusion. Given that recent CASPs have released a number of FM targets and all the CASP models and most experimental structures become available now, the authors should benchmark their method on a larger dataset (e.g., combining FM domains from CASP11-13) and compare it with the CASP servers separately, which will help draw a more meaningful and objective conclusion.

2. The major novelty (although not a big one) of the pipeline, compared to the existing pipeline by Xu, is the iterative integration of the model training and structure construction. However, the manuscript failed to show convincing evidence that the iteration does help to improve the distance and/or structure prediction. It only presents an anecdote example in Fig 6. A more systematic analysis is needed, e.g., on the data of the average accuracy of distance prediction and average TM-scores of structural models in each of the iterations. In fact, on the statement of "13 of 22 cases show lower mean absolute error at the last iteration than at the first iteration", the effect of the iteration may be quite modest. But anyhow, a more quantitative analysis is needed here.

Other concerns:

1. P4, the mean distance error of 5.6 Å is very high. This may be because the authors counted for all the distance predictions with bin <15 Å. However, it seems that DMPfold uses only the distances with some maximum likelihood cutoff (>0.4?). If it is true, the authors should probably calculate the error for the distances that are used by DMPfold; it does not make sense to calculate the error for those not used.

2. In the Pfam validation, it seems that the authors compared their models with the HHsearch templates, which is a concern to me. I think that the authors should at least add the data for which the models are compared directly with the experimental structure, even though they may

have a lower number of test proteins from Pfam.

3. Please give more details about the training of the iterative model (with the additional dimension of prior CB-CB distances); especially, how to obtain the additional feature for all training data? What is the loss function to train the torsion angles? What is the loss function to predict the reliability of predictions of torsion angles?

I believe some of the above concerns can be addressed in a revision. But my major concerns on the lack of overall novelty and the poor performance of the approach in recent CASP made me think the work is unsuitable for publication in this journal.

Reviewer #2 (Remarks to the Author):

Summary:

This paper presents an iterative method to predict inter-residue distances with deep learning and build 3D protein structures from predicted distances using the distance geometry. The iterative process is interesting. The prediction of H-bond acceptor and donor contacts is relatively new. The method improves the accuracy of ab initio structure folding on some CASP datasets in comparison with two widely used contact-based modeling methods. It was also successfully applied to folding proteins without known structures in the Pfam database and to some membrane proteins. The method is also available as an open source tool useful for the community.

Specific comments for revision:

1. How do you handle the predicted distances that are ≥ 19 angstrom in structure modeling using CNSsolve? How are their upper bounds set?
2. How does the accuracy of Cb-Cb distance prediction with deep learning change from iteration to iteration? Can you report the detailed statistical results?
3. Can you report the accuracy of prediction of H-bond donor and acceptor contacts using deep learning?
4. For the CASP12 FM domains, predictions where the bin with the maximum likelihood is less than 15 Å were compared to the true distances. The absolute errors between the bin centres and the true distances has mean 5.6Å and median 3.1 Å, indicating predictions good enough to build accurate models. This description is not very clear. Can you explain how the predicted distance for a residue pair is selected more clearly? Is it equal to the center of the bin with the maximum likelihood?
5. Figure 6(C) is used to show how the modeling is improve over iterations. The native structure (PDB ID 2L6O) is in blue, the initial model is in orange and the model after iterations is in green. However, in the highlighted red circle, the helix in orange matches the blue helix is better than the green helix, which seems to suggest the initial model is better. Can you explain this figure better?

Reviewer #3 (Remarks to the Author):

This paper describes a new method for generating the 3D structure of a protein, given a predicted distance map, and uses this method to predict the structures of a large number of Pfam domains.

It builds upon an existing method from the same lab, DeepMetaPSICOV, but extends it in important ways and assesses the value of these extensions using a cross-validation scheme.

There are two areas of novelty / value that this paper brings. First is the development of a new structure prediction method. DMPfold generalizes contact map prediction, in which the C_β atoms of two residues are predicted to either interact or not (typically with an 8 angstrom threshold), to distance prediction, by discretizing the distance between any two C_β atoms into small bins and then predicting the likelihood of a pair falling in any given bin. The method also predicts new modalities beyond a contact map, specifically a hydrogen bonding network and torsion angles, and uses all this information to then iteratively fold the protein into a 3D structure using a constraint satisfaction approach. The individual innovations are largely novel and so is their integration. The approach is similar in spirit to what AlphaFold, the best performing server in the FM category at CASP13, does, but that method is not yet published. The approach is also similar in spirit to RaptorX-DeepModeller, with some important differences, but that method too has not yet been published (it has been preprinted however). Even if these methods were published, having multiple independent approaches try out similar but meaningfully differentiated ideas is worthwhile and contributes to a better understanding of how the methods work.

The second contribution is the prediction of a large number of Pfam families. This is not novel, as the paper properly references a number of similar undertakings, but it is of value as demonstrated in Figure 4C, in which the overlap in high-quality predictions between DMPfold and previous efforts is shown to be fairly small, with a large number of new high-quality predictions (977). I believe this will be of value to the broader biology community, especially if it is exposed in a manner that is readily accessible and kept up-to-date. A nice aspect of the analysis is their use of a neural network to predict the quality of a predicted model, which is of paramount importance for broader adoption of this technology by non-specialists.

I believe this paper will influence the field in two ways. First, it demonstrates the value of adding distance information, beyond mere binary contacts, to the quality of generated folds. This is likely to become a field-wide shift. Second, it more clearly delineates the value of protein structure prediction to the broader biology community, by identifying what fraction of structures can be reliably predicted, but also what limitations remain (e.g. multi-domain proteins, proteins without Pfam annotations, etc.) Therefore, it provides a sort of status update on what value protein structure prediction currently provides and where important improvements can be made moving forward.

The fact that the authors make the code available on Github ensures that it is readily reproducible and that others can build on it.

Some important critiques / suggestions:

The way the manuscript is currently written understates the methodological innovations. Space-permitting, I suggest adding a section to the results, perhaps the very first section, that briefly summarizes the main developments in the new method vs. DeepMetaPSICOV. This information is currently in the supplement, but it would help greatly in explaining the value of the paper to put the methodological changes front and center. Currently the reader could be forgiven for thinking that the paper is primarily about applying DMPfold to Pfam, as opposed to developing the DMPfold method itself.

The performance gains of DMPfold are convincingly shown using a cross-validation scheme, but additional baselines would be helpful. In particular, repeated claims of achieving "state of the art" performance are made, but it's unclear with respect to what this claim is being established. The only external baseline used is Rosetta, which performed best at CASP12, but not at CASP13. Thus it would help to clarify that the SOTA claim is made being with respect to CASP12. Alternatively, if the comparison is meant with respect to CASP13, then the assessments should be extended to at

least include RaptorX, which is publicly available (unlike AlphaFold), and which performed better at CASP13 than DMPfold. To be sure, this could potentially be quite difficult to do as RaptorX has likely been trained on data available subsequent to CASP12, which would make the comparison unfair to DMPfold. Therefore it may be best to restrict the SOTA claim, as I don't think it's important to the overall novelty and value of this paper. The methodological innovations on their own are quite interesting.

Related to this, it is stated that this method was used at CASP13 but it's unclear under which group. DMP is present under the contact prediction category but not under the regular targets category. I'm assuming the Jones-UCL group is the one that used DMPfold but it's best to make this explicit in the paper.

Additional value would be gained from the methodological innovations if their relative contributions were assessed using ablation studies that measured the performance of the model with and without the individual innovations. In particular, it is unclear right now whether predictions of the hydrogen bond network and torsion angles are resulting in higher-quality predictions, or whether distances are sufficient to provide the bulk of the observed performance gain. Teasing this out would be valuable to the structure prediction community.

In the transmembrane prediction section, it is stated that the training set has no overlap with FILM3 dataset, but no measure of sequence similarity is provided. Some measure should be provided, unless there's no detectable homology whatsoever, in which case it's useful that this be made known.

In the methods section, it is stated that the LSTM transforms the 3D tensor $64 \times L \times L$ into a 2D tensor $256 \times L$. How is this done? Specifically, a 2D map is effectively being converted into a 1D sequence. Presumably some form of concatenation is being carried out of the 2D features to turn them into 1D features but this is not sufficiently explained in the text.

Segregation of the training set from various validation sets, including Pfam families, is done based on the ECOD T-group. It would be helpful to ground this in terms that are more readily accessible to the broader biology community, especially given the broad readership of Nature Communications. Something like expected maximum sequence identity would be useful, even if it is only a rough approximation of the amount of separation provided by ECOD T-groups.

On pg. 4, it is stated that in 13 out of 22 cases iteration improves the results. It would be helpful to show this as a graph, where the quality of each prediction is shown as a function of the number of iterations it has undergone, so the full trajectory can be seen.

On pg. 9, this sentence "This network has a precision of 82.5%...chain topology." does not sufficiently characterize the relationship between reliability scores and actual structure quality, as measured by TM-score. I suggest a scatterplot directly comparing the two quantities, with a correlation value shown. If there is not enough space in the main text figures it should at least be provided in the supplement.

The model was trained on proteins of length 500 or less, but was used to predict structures of up to length 800. Figure 5B seems to suggest that there's a precipitous drop in model accuracy for proteins longer than 500, but there are not enough data points to say conclusively. In general if this tool is meant to be used by the broader biology community, more careful assessment of the model's behavior in the >500 residue regime, or perhaps retraining with longer proteins, may be warranted. Alternatively, capping predictions at 500 residues is also a possibility.

On pg. 10, it is stated that a 200 residue protein takes about 3 hours. It would be helpful to characterize this more thoroughly by providing a scatterplot of prediction time as a function of protein length.

On the minor side:

More detailed description of hyperparameter choices in the methods would be helpful (the fact that the code is available on Github is great, and will ensure reproducibility, but it would be useful to have the key aspects described in the methods section as well so that other researchers do not have to inspect the code.)

In Figure 2 showing a scatter plot against Rosetta would be helpful, in addition to the smoothed histograms already shown in 2A. The same is true of Figure 3.

Figure 5A and B should have correlation values, and perhaps the dots should be slightly transparent to get a sense of density. Furthermore, Figure 5A seems to suggest, although not conclusively, that prediction performance does not markedly improve past having $\sim 10^3$ alignments (showing densities or something like a box and whisker chart would make the trends clearer). If this is in fact the case, it may merit some speculation in the discussion as to why, specifically why despite having very large MSAs it is still sometimes the case that only poor predictions can be made. Does this suggest a fundamental limit to this type of approach? Since DMPfold is the latest method to be applied in a systematic fashion across Pfam, it would be beneficial to the rest of the community to try to draw as many insights from the results as possible, so long as they are justified of course. This would help increase the impact of the conclusions.

The analysis of Figure 5A would also benefit from including N_{eff} in addition to number of sequences, as it may disambiguate some of the above issues.

The claim is made in pg. 15 that DMPfold works much faster than AlphaFold, but given that the latter has yet to be published, the basis for the claim should be clarified.

Some bioRxiv references are outdated (papers have been published).

Mohammed AIQuraishi

Extending genome-scale *de novo* protein modelling coverage using iterative deep learning-based prediction of structural constraints

Responses to reviewer comments

We thank the reviewers for their insightful comments that have led to improvements in the paper. We have made significant changes to the paper to address the points raised as detailed below. With these changes we believe that the manuscript is suitable for publication.

While this manuscript was under review, 9 of the Pfam families modelled at high confidence had a structure released in PDB for the first time. 8 of these 9 models have a TM-align score of at least 0.5 to the deposited structure, with a mean TM-align score of 0.62 - see Supplementary Table 4. This set acts as a further validation set for DMPfold, and the fraction of correct folds matches the precision of the model accuracy predictor. The one model with TM-align score less than 0.5 corresponded to a *de novo*-modelled region of a 4 Å resolution cryo-EM structure, which may not be reliable. We believe this extra data, which has been added to the discussion, provides further evidence for the accuracy and timeliness of this study.

Reviewers' comments and responses:

Reviewer #1 (Remarks to the Author):

This manuscript proposed to model protein structures using neural-network based residue distance prediction. It is a straightforward extension of the DeepMetaPSICOV (DMP) program, developed by the same group, but with contact map replaced by distance map. Similar work was done by Xu as cited in Ref [14]. There is a modest novelty in which the iterations are introduced between distance training and model generations. The work catches up the hot topic of distance-guided structure prediction, which started to become popular since the work of the AlphaFold team in CASP13. But there are substantial problems in the way that the authors present and benchmark their method. My major concern on this work is that it does not represent the state of the work.

Major concerns:

1. The method was only compared to Confold2 and Rosetta based on a small set of 22 CASP12 FM domains. First of all, Confold2 and Rosetta in the comparison do not use distance constraints. The authors converted their distance prediction into contacts and then used the converted contacts to guide Confold2 and Rosetta. Thus, it is not clear how much information was lost in the conversion and how accurate the converted

contacts are. Why do not you simply use DMP contact map? Second, since you are testing your method on the CASP12 domains, why do not you directly compare your models with the released models by the CASP12 servers? This will help examine how far your method is compared to the state of the art (3 years ago) Third, the time is now post CASP13 and methods in CASP13 had a significant progress compared to the approaches in CASP12. To benchmark with the real state of the art, the method must be compared to the CASP13 servers. In fact, the Jones-UCL group performed quite poor, compared to the best CASP13 server predictors (ranked as No 23 at the list of [link]). This raises some concern on the performance of this presented method. Finally, the 22 FM domains are really small to draw a solid conclusion. Given that recent CASPs have released a number of FM targets and all the CASP models and most experimental structures become available now, the authors should benchmark their method on a larger dataset (e.g., combining FM domains from CASP11-13) and compare it with the CASP servers separately, which will help draw a more meaningful and objective conclusion.

We wanted to compare a distance-based model generation method to two contact-based model generation methods using the same information, hence why we summed up distance predictions to 8 Å to obtain contact predictions. If we had used DMP then it would have been unclear whether the difference in performance was due to a difference in the model generation methods or a difference between DMP and DMPfold. In practice we find that DMPfold shows similar performance to DMP for contact prediction and so using DMP would have made little difference. This has been clarified in the methods. The similar performance can be explained by the identical training sets and very similar model architectures.

We have added a comparison to the CASP12 server models to the results as suggested, indicating the development of the field since CASP12 and the effectiveness of DMPfold: “Comparing DMPfold to the CASP12 server models indicates methodological progress in the field, and is a fair comparison as DMPfold in this case uses sequence data from the time (see the Methods). The leading servers Zhang-Server and BAKER-ROSETTASERVER both obtained TM-scores above 0.5 for 8 of the 22 FM domains when considering the top model only, compared to 11 of 22 for DMPfold.”

With regards to CASP13, an early version of DMPfold was entered for assessment, but was under constant development throughout the experiment, as is often the case with new protein structure prediction methods. According to the official assessor’s ranking (http://predictioncenter.org/casp13/doc/presentations/Assessment_FM_Abriata_DalPeraro.pdf slide 19) Jones-UCL was ranked 9th among all groups when considering FM and TBM/FM targets, which are the relevant targets here. The only server that performed better was Zhang-Server (and the closely related Quark), showing the competitive performance of the method.

We used the CASP11 FM domains to develop and parameterise DMPfold (see the Methods) so these cannot be used for assessment. We believe that the comparison to CONFOLD2 and Rosetta on the CASP12 set, the comparison to FILM3 on the FILM3 set and the comparison to PconsFam and Baker group on the large Pfam set provide enough evidence as to the effectiveness of DMPfold.

2. The major novelty (although not a big one) of the pipeline, compared to the existing pipeline by Xu, is the iterative integration of the model training and structure construction. However, the manuscript failed to show convincing evidence that the iteration does help to improve the distance and/or structure prediction. It only presents an anecdote example in Fig 6. A more systematic analysis is needed, e.g., on the data of the average accuracy of distance prediction and average TM-scores of structural models in each of the iterations. In fact, on the statement of “13 of 22 cases show lower mean absolute error at the last iteration than at the first iteration”, the effect of the iteration may be quite modest. But anyhow, a more quantitative analysis is needed here.

This concern was also raised by the other reviewers. We added two more graphs to Figure 2 showing the change in TM-score and mean absolute distance error between iterations for the CASP12 FM domains. The benefit of iterations to DMPfold is shown by the fact that 19 of 22 domains show higher TM-score at the last iteration than at the first iteration, along with 13 of 22 domains showing lower mean absolute distance error as previously stated. 3 domains move from a TM-score below 0.5 to a TM-score above 0.5 over the course of the iterations. This analysis is added to the results text and indicates the importance of iterations to DMPfold.

Other aspects of the method are also quite distinct from other approaches such as Xu's method and AlphaFold e.g. the calculation of distance bounds, rather than single distances, plus the explicit prediction of main chain hydrogen bond donor acceptor pairs.

Other concerns:

1. P4, the mean distance error of 5.6 Å is very high. This may be because the authors counted for all the distance predictions with bin <15 Å. However, it seems that DMPfold uses only the distances with some maximum likelihood cutoff (>0.4?). If it is true, the authors should probably calculate the error for the distances that are used by DMPfold; it does not make sense to calculate the error for those not used.

DMPfold uses as constraints all predicted distances where the maximum likelihood bin is not the final unbounded bin (19 Å or more). The 0.4 value is the total likelihood threshold when bins are combined to obtain the upper and lower distance bounds. This has been clarified in the methods. It should be noted that the distance error is to the center of the maximum likelihood bin, not to the nearest bound (data on the fraction of satisfied bounds is provided later in that paragraph). It is also unclear that a mean absolute distance error of 5.6 Å is very high. As shown in Figure 2D this value is skewed by a few proteins; a better measure is likely the median

of 3.1 Å. We find that this is good enough to build models with the correct fold, and certainly provides more information than binary contact prediction.

2. In the Pfam validation, it seems that the authors compared their models with the HHsearch templates, which is a concern to me. I think that the authors should at least add the data for which the models are compared directly with the experimental structure, even though they may have a lower number of test proteins from Pfam.

We use a search of the PDB against the Pfam HMM with an E-value threshold of 0.001 to find templates for the validation set. The target sequences for modelling are found for each family using hmmsearch to search the UniRef90 database with the Pfam HMM and taking the closest subsequence match by E-value. In general therefore the target sequence does not have an available experimental structure with the exact same sequence, and the cases where this is true are not enough to draw firm results from. However to address this point we assembled a subset of the validation set where the template E-value threshold was 1E-6, a stricter cutoff. This subset has 66% of models with a TM-score of at least 0.5 compared to 52% for the larger set, indicating that some incorrect predictions may be due to differences between the target sequence and the template. We have added this to the results section.

3. Please give more details about the training of the iterative model (with the additional dimension of prior CB-CB distances); especially, how to obtain the additional feature for all training data? What is the loss function to train the torsion angles? What is the loss function to predict the reliability of predictions of torsion angles?

This information has been added to the methods section in the relevant places.

I believe some of the above concerns can be addressed in a revision. But my major concerns on the lack of overall novelty and the poor performance of the approach in recent CASP made me think the work is unsuitable for publication in this journal.

We have attempted to make the revisions requested by the reviewer. However we do believe that a combination of the high quality models produced, the application to the whole dark proteome and the fast and free nature of DMPfold make it a novel approach of interest to a wide audience. The issue of performance in the latest CASP has been addressed above - DMPfold was ranked in the top 10 and below only the Zhang-Server/Quark server in terms of fully automated server predictions.

Reviewer #2 (Remarks to the Author):

Summary:

This paper presents an iterative method to predict inter-residue distances with deep learning and build 3D protein structures from predicted distances using the distance geometry. The iterative process is interesting. The prediction of H-bond acceptor and donor contacts is relatively new. The method improves the accuracy of ab initio structure folding on some CASP datasets in comparison with two widely used contact-based modeling methods. It was also successfully applied to folding proteins without known structures in the Pfam database and to some membrane proteins. The method is also available as an open source tool useful for the community.

Specific comments for revision:

1. How do you handle the predicted distances that are ≥ 19 angstrom in structure modeling using CNSsolve? How are their upper bounds set?

This has been clarified in the methods: “Bounds are not generated for cases where the maximum likelihood bin is the unbounded last bin, and the last bin is also excluded from the likelihood accretion procedure. This means that all upper bounds provided to CNSsolve are < 19 Å”.

2. How does the accuracy of Cb-Cb distance prediction with deep learning change from iteration to iteration? Can you report the detailed statistical results?

As stated in more detail in the response to Reviewer 1, this has been added to Figure 2. The change of TM-score and absolute distance error with iterations is shown and discussed in the results.

3. Can you report the accuracy of prediction of H-bond donor and acceptor contacts using deep learning?

The accuracy of DMPfold at predicting H-bonds is reported in the results for the CASP12 FM domains: “The prediction of hydrogen bonds using deep learning and use of these in model generation is a novel contribution of DMPfold. The hydrogen bond predictor is accurate, with a mean of 79% of predicted hydrogen bonds present according to DSSP across the CASP12 FM domains. These predictions take into account the directionality of the hydrogen bonds (i.e. which residue is acting as donor and which is acting as acceptor), which further helps constrain the models.” In addition the importance of hydrogen bonds to model generation is explored in the ablation study - see the response to Reviewer 3 and Supplementary Table S1.

4. For the CASP12 FM domains, predictions where the bin with the maximum likelihood is less than 15 Å were compared to the true distances. The absolute errors between the bin centres and the true distances has mean 5.6Å and median 3.1 Å, indicating predictions good enough to build accurate models. This description is not very clear. Can

you explain how the predicted distance for a residue pair is selected more clearly? Is it equal to the center of the bin with the maximum likelihood?

Yes, it is equal to the center of the bin with the maximum likelihood. This has been clarified: “For the CASP12 FM domains, predictions where the bin with the maximum likelihood is less than 15 Å were compared to the true distances. The absolute error between the centre of the bin with maximum likelihood and the true distances has mean 5.6 Å and median 3.1 Å, indicating predictions good enough to build accurate models.”

5. Figure 6(C) is used to show how the modeling is improve over iterations. The native structure (PDB ID 2L6O) is in blue, the initial model is in orange and the model after iterations is in green. However, in the highlighted red circle, the helix in orange matches the blue helix is better than the green helix, which seems to suggest the initial model is better. Can you explain this figure better?

It has been clarified in the caption to Figure 6 that the loop region indicated in red and the following helix are closer to the native structure in the prediction after iterations than the initial prediction. Part of the helix is closer in the initial model than the model after iterations as the reviewer says, however this difference is less than the other regions indicated.

Reviewer #3 (Remarks to the Author):

This paper describes a new method for generating the 3D structure of a protein, given a predicted distance map, and uses this method to predict the structures of a large number of Pfam domains. It builds upon an existing method from the same lab, DeepMetaPSICOV, but extends it in important ways and assesses the value of these extensions using a cross-validation scheme.

There are two areas of novelty / value that this paper brings. First is the development of a new structure prediction method. DMPfold generalizes contact map prediction, in which the C_beta atoms of two residues are predicted to either interact or not (typically with an 8 angstrom threshold), to distance prediction, by discretizing the distance between any two C_beta atoms into small bins and then predicting the likelihood of a pair falling in any given bin. The method also predicts new modalities beyond a contact map, specifically a hydrogen bonding network and torsion angles, and uses all this information to then iteratively fold the protein into a 3D structure using a constraint satisfaction approach. The individual innovations are largely novel and so is their integration. The approach is similar in spirit to what AlphaFold, the best performing server in the FM category at CASP13, does, but that method is not yet published. The approach is also similar in spirit to RaptorX-DeepModeller, with some important differences, but that method too has not yet been published (it has been preprinted however). Even if these methods were published, having multiple independent

approaches try out similar but meaningfully differentiated ideas is worthwhile and contributes to a better understanding of how the methods work.

The second contribution is the prediction of a large number of Pfam families. This is not novel, as the paper properly references a number of similar undertakings, but it is of value as demonstrated in Figure 4C, in which the overlap in high-quality predictions between DMPfold and previous efforts is shown to be fairly small, with a large number of new high-quality predictions (977). I believe this will be of value to the broader biology community, especially if it is exposed in a manner that is readily accessible and kept up-to-date. A nice aspect of the analysis is their use of a neural network to predict the quality of a predicted model, which is of paramount importance for broader adoption of this technology by non-specialists.

I believe this paper will influence the field in two ways. First, it demonstrates the value of adding distance information, beyond mere binary contacts, to the quality of generated folds. This is likely to become a field-wide shift. Second, it more clearly delineates the value of protein structure prediction to the broader biology community, by identifying what fraction of structures can be reliably predicted, but also what limitations remain (e.g. multi-domain proteins, proteins without Pfam annotations, etc.) Therefore, it provides a sort of status update on what value protein structure prediction currently provides and where important improvements can be made moving forward.

The fact that the authors make the code available on Github ensures that it is readily reproducible and that others can build on it.

We thank the reviewer for the thorough assessment of the impact of the work.

Some important critiques / suggestions:

The way the manuscript is currently written understates the methodological innovations. Space-permitting, I suggest adding a section to the results, perhaps the very first section, that briefly summarizes the main developments in the new method vs. DeepMetaPSICOV. This information is currently in the supplement, but it would help greatly in explaining the value of the paper to put the methodological changes front and center. Currently the reader could be forgiven for thinking that the paper is primarily about applying DMPfold to Pfam, as opposed to developing the DMPfold method itself.

The section at the end of the introduction has been expanded to highlight the differences between DMP and DMPfold and to indicate that the methodological contributions of DMPfold are an important contribution from the paper: "Here we introduce DMPfold, a development of our DeepMetaPSICOV (DMP) contact predictor [17]. Rather than predicting contacts, DMPfold predicts inter-atomic distance bounds, torsion angles and hydrogen bonds and uses these constraints to build models. An iterative process of model generation and constraint refinement

is used to filter out unsatisfied constraints. Other modifications to the neural network architectures also differentiate DMPfold from DMP - see the Methods.”

The performance gains of DMPfold are convincingly shown using a cross-validation scheme, but additional baselines would be helpful. In particular, repeated claims of achieving “state of the art” performance are made, but it’s unclear with respect to what this claim is being established. The only external baseline used is Rosetta, which performed best at CASP12, but not at CASP13. Thus it would help to clarify that the SOTA claim is made being with respect to CASP12. Alternatively, if the comparison is meant with respect to CASP13, then the assessments should be extended to at least include RaptorX, which is publicly available (unlike AlphaFold), and which performed better at CASP13 than DMPfold. To be sure, this could potentially be quite difficult to do as RaptorX has likely been trained on data available subsequent to CASP12, which would make the comparison unfair to DMPfold. Therefore it may be best to restrict the SOTA claim, as I don’t think it’s important to the overall novelty and value of this paper. The methodological innovations on their own are quite interesting.

References to “state of the art” have been removed from the final introduction paragraph and the Pfam analysis section to avoid confusion.

Related to this, it is stated that this method was used at CASP13 but it’s unclear under which group. DMP is present under the contact prediction category but not under the regular targets category. I’m assuming the Jones-UCL group is the one that used DMPfold but it’s best to make this explicit in the paper.

The group was Jones-UCL and this has been made explicit in the discussion where CASP13 is mentioned.

Additional value would be gained from the methodological innovations if their relative contributions were assessed using ablation studies that measured the performance of the model with and without the individual innovations. In particular, it is unclear right now whether predictions of the hydrogen bond network and torsion angles are resulting in higher-quality predictions, or whether distances are sufficient to provide the bulk of the observed performance gain. Teasing this out would be valuable to the structure prediction community.

To assess this we carried out an ablation study for all 8 combinations of the 3 constraint types (distance, H-bond and torsion). This is shown in Supplementary Table S1 and discussed in the results: “The importance of the three constraint types (distance, torsion and H-bond) to DMPfold is shown in Supplementary Table S1. Whilst distance constraints are required for successful structure prediction, adding torsion and H-bond constraints to distance constraints does lead to improved performance. The best performance is achieved when all three constraint types are combined. The prediction of hydrogen bonds using deep learning and use of these in model

generation is a novel contribution of DMPfold. The hydrogen bond predictor is accurate, with a mean of 79% of predicted hydrogen bonds present according to DSSP across the CASP12 FM domains. These predictions take into account the directionality of the hydrogen bonds (i.e. which residue is acting as donor and which is acting as acceptor), which further helps constrain the models.”

In the transmembrane prediction section, it is stated that the training set has no overlap with FILM3 dataset, but no measure of sequence similarity is provided. Some measure should be provided, unless there's no detectable homology whatsoever, in which case it's useful that this be made known.

It has been made clear in the text that there is no overlap at the ECOD T-group level between the training set and the FILM3 dataset. See below response to the comment about ECOD for more on this.

In the methods section, it is stated that the LSTM transforms the 3D tensor $64 \times L \times L$ into a 2D tensor $256 \times L$. How is this done? Specifically, a 2D map is effectively being converted into a 1D sequence. Presumably some form of concatenation is being carried out of the 2D features to turn them into 1D features but this is not sufficiently explained in the text.

This has been clarified in the methods: “Torsion angle predictions are generated with the same deep residual maxout network, but with the 20-dimensional softmax output layer replaced by a bidirectional recurrent LSTM layer [35] with 128 hidden units (BLSTM in Figure 9C), which embeds each row of the final 2-D 64-channel feature map in a single 256-D vector (concatenation of 128-D final timestep output states of the forward and reverse direction LSTM passes). As each row ends up embedded in a single vector, the LSTM layer thus transforms the 3-D tensor (64 features \times L \times L) into a 2-D tensor (256 features \times L), which is finally reduced in dimensions to three final feature channels (3 \times L), interpreted as ϕ , ψ and ω angles in radians for each residue.”

Segregation of the training set from various validation sets, including Pfam families, is done based on the ECOD T-group. It would be helpful to ground this in terms that are more readily accessible to the broader biology community, especially given the broad readership of Nature Communications. Something like expected maximum sequence identity would be useful, even if it is only a rough approximation of the amount of separation provided by ECOD T-groups.

We have added a paragraph to the methods explaining why we use ECOD T-group rather than sequence identity to split our datasets: “In order to assess the generalisation ability of any trained deep learning model, it is crucial to ensure that the set of examples used to train the model does not share any obvious overlap with examples used to test the model. For proteins, a criterion based on sequence identity between any training and test example is commonly used.

However, this is generally insufficient to rule out an evolutionary relationship or structural similarity, as many related proteins share less than 20% sequence identity. It is much more preferable to use a structural classification database such as CATH, SCOPe, or ECOD. We use the evolutionary classification of domains (ECOD) database [21] to define our train-test split for all benchmarks in this paper.” More information can be found in papers describing ECOD.

On pg. 4, it is stated that in 13 out of 22 cases iteration improves the results. It would be helpful to show this as a graph, where the quality of each prediction is shown as a function of the number of iterations it has undergone, so the full trajectory can be seen.

As stated in more detail in the response to Reviewer 1, this has been added to Figure 2 and shows the change of TM-score and absolute distance error with iterations.

On pg. 9, this sentence “This network has a precision of 82.5%...chain topology.” does not sufficiently characterize the relationship between reliability scores and actual structure quality. as measured by TM-score. I suggest a scatterplot directly comparing the two quantities, with a correlation value shown. If there is not enough space in the main text figures it should at least be provided in the supplement.

The scatter plot comparing predicted and actual model quality has been added as Supplementary Figure S2, along with the correlation value. The correlation value is mentioned in the main text.

The model was trained on proteins of length 500 or less, but was used to predict structures of up to length 800. Figure 5B seems to suggest that there’s a precipitous drop in model accuracy for proteins longer than 500, but there are not enough data points to say conclusively. In general if this tool is meant to be used by the broader biology community, more careful assessment of the model’s behavior in the >500 residue regime, or perhaps retraining with longer proteins, may be warranted. Alternatively, capping predictions at 500 residues is also a possibility.

It is correct that there are not enough proteins over 500 residues to draw firm conclusions, though it does appear there is a drop off in model prediction accuracy over 500-600 residues. We have expanded the paragraph discussing this in the results: “Beyond this, the accuracy falls with the default parameters used. This dropoff in accuracy may also stem from the DMP training set, which had a maximum chain length of 500 residues. In addition, some of these longer proteins have multiple structural domains despite being from one Pfam family, which can make modelling and assessment hard. Mirror topology effects can also be an issue - see the Discussion. We recommend that users treat DMPfold models for proteins of more than 500 residues with caution and consider splitting them up.”

On pg. 10, it is stated that a 200 residue protein takes about 3 hours. It would be helpful to characterize this more thoroughly by providing a scatterplot of prediction time as a function of protein length.

A new figure, Supplementary Figure S1, has been added to show the scatter plot of run time as a function of protein length for the CASP12 domains.

On the minor side:

More detailed description of hyperparameter choices in the methods would be helpful (the fact that the code is available on Github is great, and will ensure reproducibility, but it would be useful to have the key aspects described in the methods section as well so that other researchers do not have to inspect the code.)

Details on hyperparameters and training have been added to the methods: “Training of all models was performed using the Adam optimizer [45] for 75 epochs with default parameters ($\beta_1=0.9$, $\beta_2=0.999$, maximum learning rate of 10^{-3}), the final model weights were those that produced the lowest cross entropy loss on the validation set (5% of the training cases held out). Minibatches of size one were used for forward and backward passes due to GPU memory limitations, although gradients were accumulated and averaged across minibatches of size 8. All other aspects of training, including data augmentation procedures was carried out as previously described for DMP [17].”

In Figure 2 showing a scatter plot against Rosetta would be helpful, in addition to the smoothed histograms already shown in 2A. The same is true of Figure 3.

The scatter plot comparing to Rosetta was added in Figure 2. In Figure 3 the scatter plot comparing to the FILM3 paper results was added.

Figure 5A and B should have correlation values, and perhaps the dots should be slightly transparent to get a sense of density. Furthermore, Figure 5A seems to suggest, although not conclusively, that prediction performance does not markedly improve past having $\sim 10^3$ alignments (showing densities or something like a box and whisker chart would make the trends clearer). If this is in fact the case, it may merit some speculation in the discussion as to why, specifically why despite having very large MSAs it is still sometimes the case that only poor predictions can be made. Does this suggest a fundamental limit to this type of approach? Since DMPfold is the latest method to be applied in a systematic fashion across Pfam, it would be beneficial to the rest of the community to try to draw as many insights from the results as possible, so long as they are justified of course. This would help increase the impact of the conclusions.

Correlation values have been added to Figures 5A and 5B and the dots have been made transparent to make the plots clearer. The results showing accuracy for each alignment depth

range are also expanded in the results: “When the alignment contains 50-100 sequences the chance of generating a model with the correct fold is 22%. This rises to 38% for alignments with 100-200 sequences, 57% for 200-500 sequences, 58% for 500-1000 sequences, 66% for 10^3 - 10^4 sequences and 84% for 10^4 or more sequences.” So it does appear that increasing the alignment size appears to give more accurate models even after $\sim 10^3$ alignments.

The analysis of Figure 5A would also benefit from including N_{eff} in addition to number of sequences, as it may disambiguate some of the above issues.

This has been added as a new plot in Figure 5A.

The claim is made in pg. 15 that DMPfold works much faster than AlphaFold, but given that the latter has yet to be published, the basis for the claim should be clarified.

A citation to the CASP13 presentation by John Jumper, which included a plot of computation time, has been added. This is available on the CASP website. If the AlphaFold paper is published before this paper the citation will be updated accordingly.

Some bioRxiv references are outdated (papers have been published).

These have been updated.

Mohammed AlQuraishi

REVIEWERS' COMMENTS:

Reviewer #1 (Remarks to the Author):

(M1). Authors argued that "If we had used DMP then it would have been unclear whether the difference in performance was due to a difference in the model generation methods or a difference between DMP and DMPfold". This is exactly what my concern was. You claimed that DMPfold outperforms Rosetta and Confold2 in your test. Then it is not clear if the difference in performance was due to model generation methods or a difference between distance and contact predictions. If you really want to demonstrate the advantage of distance predictions, you should compare DMPfold with distance and DMPfold with contact constraints.

(M2). The new data on TM-score is quite confusing to me. In previous data, 13 out of 22 domains showed lower distance error, meaning that the iterations reduced the modeling accuracy in 9 cases. However, in the TM-score data there are only 3 cases in which TM-score is reduced.

(O1). It is not very clear how the distance constraints are used in the description. Do you mean that you used all the distance constraints but with the width of intervals between upper and lower bounds changed according to the confidence level of distance predictions? Have you tried to optimize the procedure, e.g., using only high confidence distance predictions? The outliers with large distance errors in Fig 2D might be from the very low-confidence distance predictions.

(O2). I am still not convinced with the stringency of using predicted models (by HHsearch) to assess predicted models (by DMPfold). No matter what cutoffs you use, there is still a chance that the predicted model is wrong.

Overall, I found that the conclusion of my last round of review, "I believe some of the above concerns can be addressed in a revision. But my major concerns on the lack of overall novelty and the poor performance of the approach in recent CASP made me think the work is unsuitable for publication in this journal", still hold. The authors argued that "DMPfold was ranked in the top 10 and below only the Zhang-Server/Quark server in terms of fully automated server predictions." Apparently, it is not fair to compare a human group (which I believe Jones-UCL was) with other server groups.

Reviewer #2 (Remarks to the Author):

The authors addressed my comments well.

Reviewer #3 (Remarks to the Author):

The authors have addressed all my concerns. I consider the manuscript ready for publication.

Mohammed AlQuraishi

Deep learning extends de novo protein modelling coverage of genomes using iteratively predicted structural constraints

Responses to reviewer comments

We thank the reviewers for their constructive comments. With these changes we believe that the manuscript is ready for publication.

Reviewers' comments and responses:

Reviewer #1 (Remarks to the Author):

(M1). Authors argued that “If we had used DMP then it would have been unclear whether the difference in performance was due to a difference in the model generation methods or a difference between DMP and DMPfold”. This is exactly what my concern was. You claimed that DMPfold outperforms Rosetta and Confold2 in your test. Then it is not clear if the difference in performance was due to model generation methods or a difference between distance and contact predictions. If you really want to demonstrate the advantage of distance predictions, you should compare DMPfold with distance and DMPfold with contact constraints.

We have added results comparing DMPfold with distance and DMPfold with contact constraints. "The benefits of using distance constraints rather than contact constraints were also examined. DMPfold was run with 8 Angstrom constraints for residue pairs with a cumulative likelihood of at least 0.5 for bins up to 8 Angstrom. Other aspects such as the iterations, torsion constraints and H-bond constraints were not changed. In this case the TM-scores have mean 0.43, median 0.43 and 10 domains have TM-score above 0.5. By comparison to Table 1 it can be seen that using distances produces better models than using contacts alone."

(M2). The new data on TM-score is quite confusing to me. In previous data, 13 out of 22 domains showed lower distance error, meaning that the iterations reduced the modeling accuracy in 9 cases. However, in the TM-score data there are only 3 cases in which TM-score is reduced.

13 of 22 domains show lower distance error and 19 of 22 domains show better TM-score over the iterations. These results are shown in Figure 2D. From that figure it can be seen that some of the changes are marginal, which accounts for some of the difference. In addition, most of the cases of distance error increasing occur when the model has a TM-score below 0.5. When the model is incorrect and the distance error is 7 Angstrom or more the relationship between a small change in distance error and a small change in TM-score is unclear, which also accounts for the difference.

(O1). It is not very clear how the distance constraints are used in the description. Do you mean that you used all the distance constraints but with the width of intervals between upper and lower bounds changed according to the confidence level of distance predictions? Have you tried to optimize the procedure, e.g., using only high confidence distance predictions? The outliers with large distance errors in Fig 2D might be from the very low-confidence distance predictions.

Distance constraints are used for modelling in all cases where the maximum likelihood bin is not the unbounded last bin. The width of the intervals is determined by the procedure outlined in the methods, i.e. growing out from the maximum likelihood bin until the total likelihood reaches 0.4.

This means that lower confidence distance predictions will have wider bounds, as more bins will be included before the threshold of 0.4 is reached. We did change the value of 0.4 but found that model quality was relatively insensitive to the value.

(O2). I am still not convinced with the stringency of using predicted models (by HHsearch) to assess predicted models (by DMPfold). No matter what cutoffs you use, there is still a chance that the predicted model is wrong.

Despite the changes to add a more stringent sub-group there is still a chance the templates used for comparison are wrong. If this gave us an advantage it would be more of an issue; given that it suggests a slight underestimation of the accuracy of DMPfold, we think this is acceptable. We think it is more important to have a procedure for selecting target sequences that is consistent across families with and without structure than to select a target sequence corresponding exactly to a PDB entry.

Overall, I found that the conclusion of my last round of review, “I believe some of the above concerns can be addressed in a revision. But my major concerns on the lack of overall novelty and the poor performance of the approach in recent CASP made me think the work is unsuitable for publication in this journal”, still hold. The authors argued that “DMPfold was ranked in the top 10 and below only the Zhang-Server/Quark server in terms of fully automated server predictions.” Apparently, it is not fair to compare a human group (which I believe Jones-UCL was) with other server groups.

Our response to this remains the same as given previously. The comparison of the Jones-UCL human group to the server groups was suggested by the reviewer in the first review round.

Reviewer #2 (Remarks to the Author):

The authors addressed my comments well.

Reviewer #3 (Remarks to the Author):

The authors have addressed all my concerns. I consider the manuscript ready for publication.

Mohammed AlQuraishi